https://doi.org/10.1038/s42003-020-1076-0　　**OPEN**

# Reciprocal H3.3 gene editing identifies K27M and G34R mechanisms in pediatric glioma including NOTCH signaling

Kuang-Yui Chen [1,2], Kelly Bush[1,2], Rachel Herndon Klein[1,2], Vanessa Cervantes[1,2], Nichole Lewis[1,2], Aasim Naqvi [2], Angel M. Carcaboso [3], Mirna Lechpammer[4] & Paul S. Knoepfler [1,2,5 ✉]

Histone H3.3 mutations are a hallmark of pediatric gliomas, but their core oncogenic mechanisms are not well-defined. To identify major effectors, we used CRISPR-Cas9 to introduce H3.3K27M and G34R mutations into previously H3.3-wildtype brain cells, while in parallel reverting the mutations in glioma cells back to wildtype. ChIP-seq analysis broadly linked K27M to altered H3K27me3 activity including within super-enhancers, which exhibited perturbed transcriptional function. This was largely independent of H3.3 DNA binding. The K27M and G34R mutations induced several of the same pathways suggesting key shared oncogenic mechanisms including activation of neurogenesis and NOTCH pathway genes. H3.3 mutant gliomas are also particularly sensitive to NOTCH pathway gene knockdown and drug inhibition, reducing their viability in culture. Reciprocal editing of cells generally produced reciprocal effects on tumorgenicity in xenograft assays. Overall, our findings define common and distinct K27M and G34R oncogenic mechanisms, including potentially targetable pathways.

[1] Department of Cell Biology and Human Anatomy, University of California, Davis, CA 95616, USA. [2] Institute of Pediatric Regenerative Medicine, Shriners Hospital for Children Northern California, Sacramento, CA 95817, USA. [3] Institut de Recerca Sant Joan de Déu, Barcelona, Spain. [4] Genome Center, University of California, Davis, CA 95616, USA. [5] Department Pathology and Laboratory Medicine, University of California, Davis, CA, USA. ✉email: knoepfler@ucdavis.edu

Pediatric high-grade gliomas are a major cause of childhood cancer deaths. Commonly recurring mutations in histone variant H3.3 (K27M and G34R/V) distinguish pediatric from adult gliomas[1–5]. Studies of the H3.3K27M mutation implicate perturbation of genomic H3K27me3 as a potential driver of gliomagenesis, including in diffuse intrinsic pontine glioma (DIPG) and glioblastoma (GBM)[6–9], but other unique features have also been identified[10–12]. GBMs bearing G34R mutations exhibit genome-wide changes in H3K36me3 profiles and elevated *MYCN*[13]. More recently, increased expression of NOTCH pathway genes was implicated in some DIPGs and NOTCH path inhibition contributed to reduced tumor growth in culture[14], but the specificity of these treatments for H3.3 mutant cells remains unclear. Pediatric glioma prognosis remains dismal, with a nearly 100% mortality rate.

In studies using exogenous expression plasmids[15,16], transgenic mouse models, or xenografts[17–19], the H3.3K27M mutation consistently depended on synergy with additional oncogenic alterations to promote tumorigenesis. A mouse model study implicated alterations in bivalent genes in mutant H3.3 downstream function[20]. More recently, CRISPR-mediated knockout of the mutant *H3F3A* allele in two H3.3K27M glioma lines restored more normal histone mark deposition and reduced tumorigenesis[9].

Here we report successful gene editing of H3.3K27M and G34R point mutations in human normal brain and pediatric glioma cells, generating pairs of otherwise isogenic cell lines that allowed us to define epigenetic and transcriptomic changes contributing to gliomagenesis in the native genomic context. Strikingly, K27M and G34R induced many overlapping changes, including reduced overall H3K27me3 and induction of NOTCH pathway genes, including *ASCL1*. In K27M lines, the decrease of H3K27me3 was especially pronounced at super-enhancers of specific gene clusters, including the NOTCH pathway genes. H3.3 mutant cells exhibited significant sensitivity to NOTCH pathway inhibition and knockdown, resulting in decreased cell viability under treatment. Glioma lines that were edited to correct the pathogenic *H3F3A* mutation exhibited less invasiveness and growth in xenografted mice than their mutated counterparts. Taken together, our findings suggest striking overlap between putative mechanisms of K27M and G34R mutations and point to increased NOTCH signaling playing a key role in K27M-related gliomagenesis, resulting from aberrant gene derepression due to impaired H3K27me3 deposition at super-enhancers in H3.3 mutant cells.

## Results

**CRISPR gene editing of H3.3 mutations in pediatric gliomas.** To study the transformative effects of oncohistone mutations in human pediatric gliomas in an otherwise isogenic context, we used CRISPR-Cas9 gene editing to revert H3.3K27M mutations in glioma cells back to wild type (WT) while in parallel introducing H3.3K27M and G34R point mutations into human astrocytes and H3.3WT glioma cells (Fig. 1a). The rationale for developing this model system was that identification of epigenetic and signaling pathways that are altered in a reciprocal manner by the introduction or reversion of these mutations would allow us to identify universal and distinct transformative mechanisms of H3.3K27M and G34R mutations.

We assembled an initial panel of cell lines as candidates for gene editing. These included eight H3.3K27M pediatric DIPG cell lines, one H3.3WT pediatric GBM line, and an H3.3WT human astrocyte (HA) line as normal cell control (Supplementary Fig. 1a). We observed the growth and phenotypes of the cell lines in culture and assayed for H3.3 and H3 histone mark levels

across this cell line panel (Supplementary Fig. 1b–d, Supplementary Table 1, Supplementary Fig. 8). We ultimately selected both H3.3WT lines and the two H3.3K27M DIPG lines (SU-DIPG-XIII and SU-DIPG-XVII, hereafter referred to as XIII and XVII, respectively[17,21]) for gene editing.

Following CRISPR-Cas9 introduction via plasmid transfection, we selected, screened, subcloned, and validated the cell lines to obtain near-clonal gene-edited lines (Supplementary Fig. 2). In this way, XIII and XVII were back-mutated from H3.3K27M to WT and are referred to as XIII-WT and XVII-WT, respectively. In addition, control cell lines (HA and SF188) that started as H3.3WT were each gene-edited to be either H3.3K27M or G34R, generating HA-K27M, HA-G34R, SF-K27M, and SF-G34R lines. Sanger sequencing of Topo-TA clones of these variants confirmed the K27M and G34R mutations in the subcloned cells (Supplementary Fig. 2c, d). We found no evidence of off-target mutations or on-target Indels and no persistent Cas9 expression (see "Methods," Supplementary Fig. 3).

**K27M and G34R alter histone marks and cell growth properties.** Parental and gene-edited cells were assayed for differences in cell growth and morphology in vitro (Fig. 1b–g). HA-G34R and XVII-WT lines exhibited reciprocal changes in proliferation rate, where introduction of the G34R mutation led to increased growth of HA-G34R compared to HA (Fig. 1b) and back mutation of K27M to WT in XVII-WT resulted in reduced proliferation compared to XVII (Fig. 1e). No significant differences in proliferation were found in the other gene-edited cell lines (Fig. 1c, d). Notably, XIII-WT and XVII-WT cells also exhibited changes in cell morphology and adherence, becoming more elongated and adherent, even when cultured in suspension conditions (Fig. 1f, g).

Introduction of K27M increased global H3.3 levels while reducing H3K27me3 levels relative to WT controls (Supplementary Fig. 4a, b), trends that we also observed across our initial panel of eight H3.3 mutant glioma lines (Supplementary Fig. 1, Supplementary Table 1, Supplementary Fig. 8). H3K27ac and H3K36me3 tracked less consistently with H3.3 mutation status (Supplementary Fig. 4a, c, d), with a mixture of significant and non-significant increases in H3K27ac with introduction of either H3.3 mutation into SF and HA cells. H3K36me3, previously linked to H3.3G34R in GBM[6], was only significantly reduced in HA-K27M compared to parental cells (Supplementary Fig. 4a, d). In addition, *MYC* and *MYCN* levels were increased in K27M-gene-edited lines and to a lesser extent in G34R-gene-edited lines (Supplementary Table 1). In our gene-edited cells, these trends of global reduction of H3K27me3, some increases in H3K27ac, and elevated *MYC* family genes in H3.3 mutant cells are mostly consistent with trends previously observed when comparing H3.3K27M-mutated gliomas to H3.3WT cells[6,9,22,23] but suggest K27M and G34R mutations may be functionally more similar than previously realized.

**H3.3 chromatin immunoprecipitation–sequencing (ChIP-seq) defines effects of K27M on H3.3 DNA binding.** We performed ChIP-seq for H3K27me3 and H3.3 on XIII, XIII-WT, XVII, and XVII-WT cells to determine the effects of K27M mutation on the distributions of H3K27me3 and H3.3 in the genome. By doing these in tandem, we could also determine whether changes in H3K27me3 were directly related to H3.3 genomic binding.

The ChIP-seq experiments were performed in biological duplicate for each cell type and antibody. After peak calling, the R package DiffBind was used to identify peak regions with differential H3K27me3 or H3.3 signal between the WT and K27M cells. In first analyzing the H3.3 ChIP-seq data, we

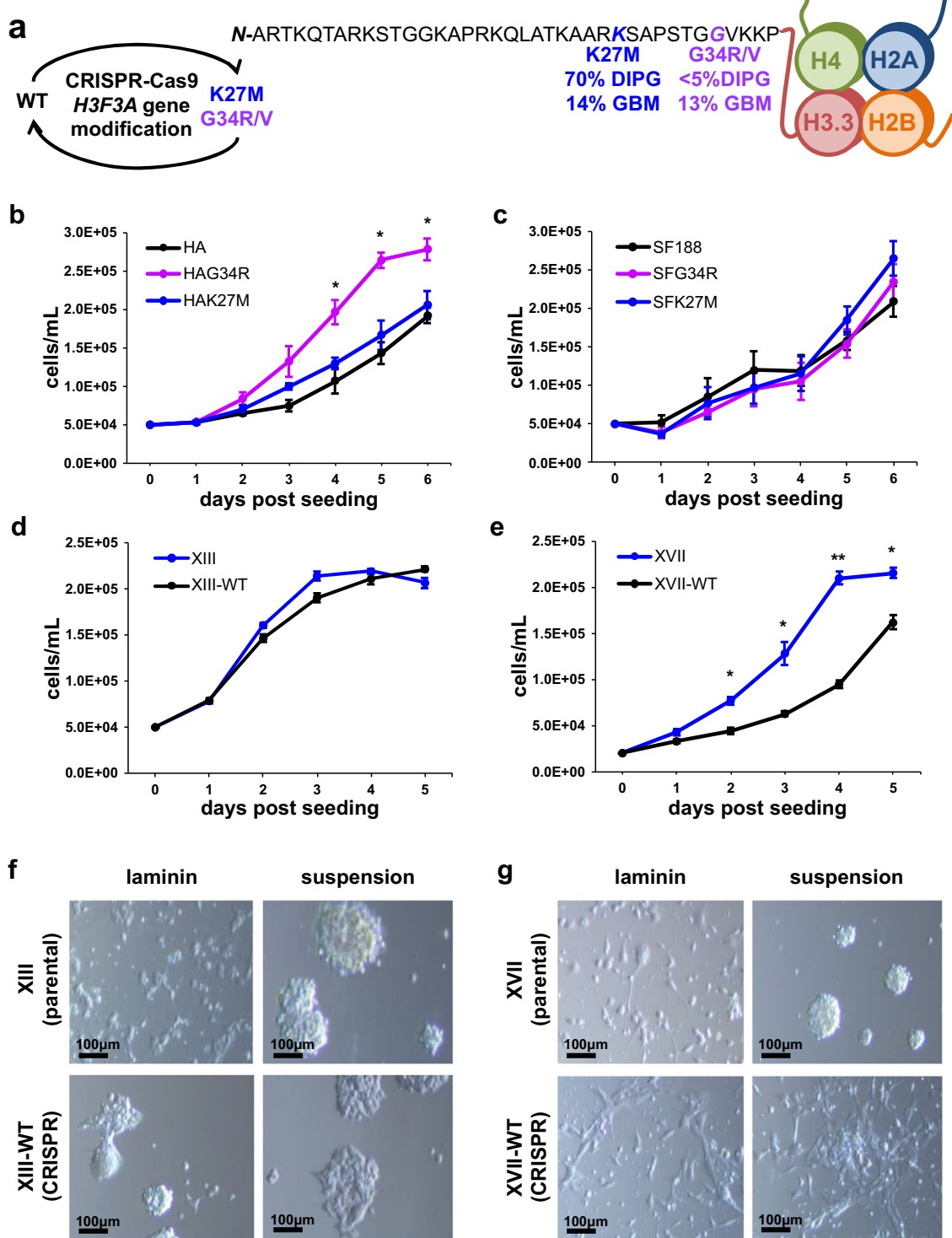

**Fig. 1 Reciprocal gene editing of histone H3.3 in pediatric glioma and control cells. a** Locations and prevalences of the most common H3.3 mutations in pediatric diffuse intrinsic pontine glioma (DIPG) and glioblastoma (GBM). **b**, **c** Proliferation rates of HA and SF188 H3.3WT cells (black) and CRISPR-mutated K27M (blue) and G34R (purple) cells were quantified over a period of 6 days. **d**, **e** Proliferation rates of XIII and XVII H3.3K27M glioma (blue) and CRISPR-reverted H3.3WT glioma (black) were quantified over a period of 5 days. XIII and XVII cells reached confluency by Day 4 and HA-G34R by Day 5. Proliferation is plotted as mean ± sd with $n = 3$ biologically independent samples for **b**, **d**, **e** and $n = 6$ biologically independent samples for **c** (*$p < 0.05$, **$p < 0.01$). **f**, **g** Brightfield microscopic images of XIII and XVII mock-transfected parental cells (unedited) and XIII and XVII H3.3WT cells (CRISPR-edited) upon growth on laminin-coated plates or in suspension culture (using low-binding suspension culture plates), as indicated in each case.

combined the data for lines XIII and XVII to compare H3.3 levels in WT and K27M cells, finding almost five times as many unique H3.3 peaks in WT cells as compared to unique peaks in isogenic K27M cells (Supplementary Table 2), suggesting that the presence of K27M perturbs normal patterns of H3.3 binding overall. Principal component analysis (PCA) shows the WT and mutant samples separate from each other along both axes but that within the WT or mutant classifications the XIII and XVII samples

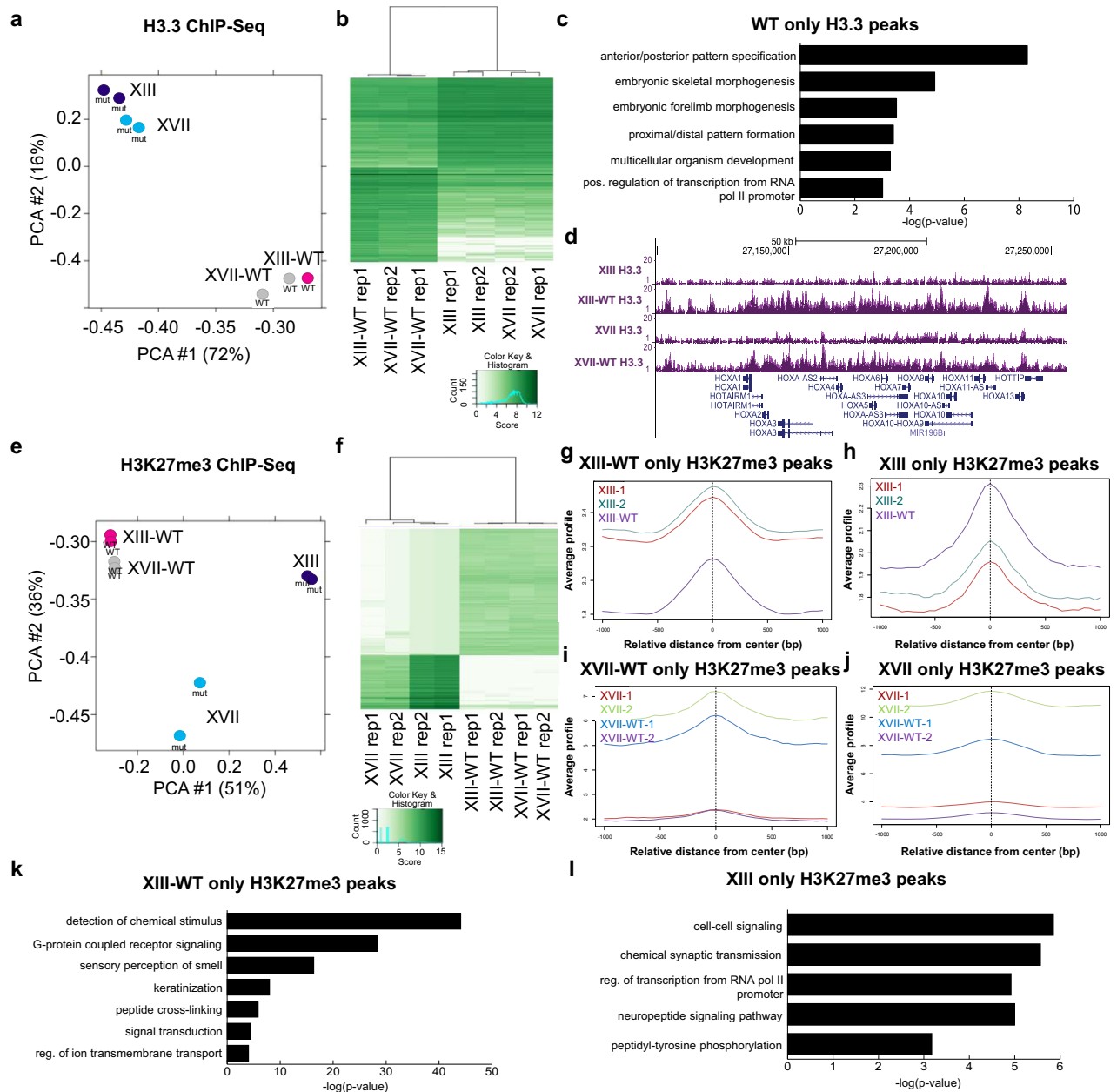

**Fig. 2 H3.3 and H3K27me3 ChIP-seq analysis in parental and gene-edited glioma lines XIII and XVII. a** Principal component (PCA) and (**b**) hierarchical clustering analysis of H3.3 ChIP-seq samples after outlier removal. **c** Gene ontology analysis of genes linked to H3.3 peaks with significant differential binding between H3.3WT cells and K27M cells (as identified by the R package DiffBind). **d** Read pileup view of H3.3 ChIP-seq data at HOXA locus, a representative region showing typical extent of gene-editing-related changes in H3.3 occupancy in XIII, XIII-WT, XVII, and XVII-WT. One of the two similar replicates is shown per cell line. **e** Principal component (PCA) and **f** hierarchical clustering analysis of biological duplicate H3K27me3 ChIP-seq samples. **g, h** H3.3 ChIP-seq signal at WT and mutant-only H3K27me3 peaks in XIII cells. **i, j** H3.3 ChIP-seq signal at WT and mutant-only H3K27me3 peaks in XVII cells, respectively. **k, l** Gene ontology analysis of genes linked to H3K27me3 peaks with significant differential binding between XIII-WT or to XIII as identified by the R package DiffBind.

cluster together (Fig. 2a). Hierarchical clustering of differential peaks further confirmed the clustering of samples based on WT or H3.3 mutant status (Fig. 2b). Motif analysis using MEME on the top 1000 H3.3 peaks most enriched in WT or H3.3 mutant cells identified several transcription factor sites, including PITX2 and regulatory factor X family members (Supplementary Table 3, Supplementary Data 1).

Gene ontology (GO) analysis of genes linked to H3.3 peaks with significantly higher signal in WT cells revealed enrichment

for categories related to embryonic development and embryo patterning, as well as transcriptional regulation (Fig. 2c). Prominent among the developmental categories were the *HOXA*, *HOXB*, and *HOXC* gene clusters (see *HOXA* data in Fig. 2d). There were no significantly enriched GO terms for genes linked to H3.3 peaks that were significantly higher in mutant H3.3 cells, fitting with the notion of H3.3K27M protein only relatively infrequently gaining direct, DNA-binding-dependent effects on gene expression as compared to WT H3.3.

**K27M drives H3K27me3 changes including in super-enhancers**. From ChIP-seq for H3K27me3 in isogenic pairs of XIII cells, 87,758 and 21,010 regions were identified as having significantly higher H3K27me3 in WT and K27M cells, respectively, when compared to each other (Supplementary Table 2). In the isogenic pairs of XVII, 27,049 and 5,190 regions had significantly higher H3K27me3 in WT and K27M cells, respectively, (Supplementary Table 2). Thus CRISPR reversion of K27M to WT consistently increases overall H3K27me3 abundance and suggest that substantially more H3K27me3 peaks are lost than are gained when the K27M mutation arises in DIPG. PCA of the H3K27me3 datasets revealed the line XIII-WT and XVII-WT samples clustering together away from the K27M mutant lines (Fig. 2e). Hierarchical clustering of the differential peaks also highlights the strong relationship of H3.3 mutation status to H3K27me3, showing sample clustering based on their status as WT or K27M rather than by cell line of origin (XIII or XVII) (Fig. 2f). Only a small fraction of regions with differential H3K27me3 between WT and mutant cells overlap regions with differential H3.3 binding (Supplementary Table 4, Fig. 2g–j), notably suggesting primarily an indirect effect of H3.3 mutation on genomic H3K27me3. Fitting with this, examining the distribution of H3.3 and H3K27me3 peaks revealed that H3.3 peaks are primarily found within gene bodies, whereas H3K27me3 peaks are more often localized to intergenic regions (Supplementary Fig. 5a). De novo motif analysis identified enriched motifs within H3K27me3 peaks (Supplementary Table 3, Supplementary Data 1).

Unexpectedly, GO analysis of the H3K27me3 peaks with significantly higher signal in XIII-WT or XVII-WT cells compared to H3.3 mutant cells revealed enrichment of some categories of genes with little apparent relation to DIPG cell function, like sensory perception of smell and keratinization (Fig. 2k, Supplementary Fig. 5b). In addition, categories related to ion transmembrane transport are enriched in these WT-specific H3K27me3 peaks (Fig. 2k, Supplementary Fig. 5b); potassium-gated ion channels were previously identified as being expressed in DIPG[17]. While less statistically significant, GO analysis for H3K27me3 peaks that are significantly higher in K27M cells included genes involved in cell signaling, synaptic transmission, and cell motility (Fig. 2l, Supplementary Fig. 5b).

Since super-enhancers have been identified in various histone H3K27M DIPG cell lines[17], we examined whether any of our peaks with differential enrichment of H3K27me3 related to H3.3 mutation status overlapped with super-enhancers. Interestingly, we identified a strong overlap of regions previously marked as super-enhancers in XIII cells[17] with regions that acquire high levels of H3K27me3 upon reversion of the K27M mutation to WT in our XIII ChIP-seq data (Fig. 3a). Substantial overlap was also observed between XIII super-enhancers and our line XVII-WT H3K27me3 peaks and between line SU-DIPG-VI super-enhancers and our XIII-WT and XVII-WT H3K27me3 peaks (Fig. 3b, c), but there was less overlap with SU-DIPG-XVII super-enhancers (Fig. 3d). Approximately a third of all super-enhancers previously identified in XIII cells gained H3K27me3 when H3.3 was reverted to WT in our study (Fig. 3a), a highly statistically significant overlap, suggesting that these regions may be aberrantly activated during DIPG tumorigenesis specifically due to K27M. Conversely, we found <1% overlap of our defined regions of altered H3K27me3 with typical enhancer regions. A number of the aberrant glioma super-enhancers functionally are linked to genes involved in NOTCH signaling including *ASCL1*, *HES5*, and *NOTCH1* (Fig. 3e–g, Supplementary Data 2) that we also identified later in this study as having increased expression in mutant cells (Supplementary Table 5), further underscoring the

link, perhaps mediated in part by super-enhancers, between H3.3 mutation and NOTCH signaling.

To validate the differential H3K27me3 abundance at super-enhancer regions observed in our ChIP-seq and compare to H3K27ac levels, we performed H3K27me3 and H3K27ac ChIP reverse transcription quantitative polymerase chain reaction (RT-qPCR) (qChIP) at ten super-enhancer regions identified via ChIP-seq as exhibiting differential H3K27me3 levels related to H3.3 mutation status, associated with the following genes: *ASCL1*, *HES5*, *BAI1*, *OLIG1*, *PDEA*, *LINGO1*, *KCNH2*, *OLIG2*, *COL20A1*, *RUNX3*. Overall across the majority of these genes, we confirmed that WT reversion of H3.3 leads to increased H3K27me3 levels in the super-enhancer regions as well as decreased H3K27ac levels in many of these same regions (Fig. 3e). Increased H3K27me3 peaks at *ASCL1* with reversion of glioma cells to WT (Fig. 3f) are representative of the changes seen at the other affected super-enhancer-linked genes. As further validation, we performed qChIP with spiked-in Drosophila chromatin for the above-mentioned genes, finding that standard and spike-in normalized qChIP exhibit almost identical large-scale changes in H3K27me3 and H3K27ac (Supplementary Fig. 6a, b).

**H3.3 mutations cause widespread gene expression changes**. We performed 3′-Tag-seq in biological duplicate on the full panel of cells to define transcriptomic changes linked to H3.3 mutations. A multiple dimensional scaling (MDS) plot of expression across all the cell lines indicates that the XIII and XVII parental cells are transcriptomically similar, while the H3.3WT HA and SF188 samples cluster opposite of the mutant gliomas (Fig. 4a). Each set of biological replicate samples clustered together, suggesting that the 3′-Tag-seq was robust. Upon gene editing, SF-K27M/G34R and HA-K27M/G34R cells′ global transcriptomic profiles shift toward that of XIII and XVII gliomas, while reversion of K27M to WT in formerly mutant glioma cells causes a substantial shift toward the HA and SF188 lines (Fig. 4a). G34R and K27M samples tended to cluster together for each set of HA and SF cells. Overall, the MDS plot highlights an axis of transformation from H3.3WT cells to H3.3 mutant gliomas and suggests partially overlapping K27M and G34R transcriptomic impacts.

Hierarchal clustering of the top 200 differentially expressed genes across all samples demonstrated clear patterns of transcriptomic shifts between parental and H3.3 gene-edited cells (Fig. 4b). We overlapped the differential expression data across all our pairs of cell lines and looked for genes that are reciprocally changed when K27M is introduced versus reverted to WT. While 139 genes were significantly higher across all K27M lines, only 6 genes were lower across K27M lines versus their WT counterparts (Fig. 4d, e), suggesting that K27M predominantly results in abnormal gene induction. GO analysis revealed gene clusters overexpressed in K27M cells including neural development (Fig. 4f). RT-qPCR validated expression changes of a select set of eight genes exhibiting the largest expression changes in the NOTCH and neurogenesis clusters: *DCX*, *GRIA4*, *RAP1GAP*, *ATCAY*, *ASCL1*, *HES5*, *DNER*, and *NOTCH1* (Fig. 4c, Supplementary Table 1). Two of these genes, *ASCL1* and *HES5*, both exhibited elevated expression and also have reduced H3K27me3 binding at their super-enhancers in H3.3 mutant cells (Figs. 3e–f and 4c Supplementary Table 5). Taken together, this analysis suggests a mechanism of oncogenic transcriptional dysregulation caused by mutant H3.3 reducing H3K27me3 at specific genes including in the case of K27M at super-enhancers.

**G34R and K27M induce many of the same genes**. Since K27M and G34R cells clustered together in the MDS analysis, we more

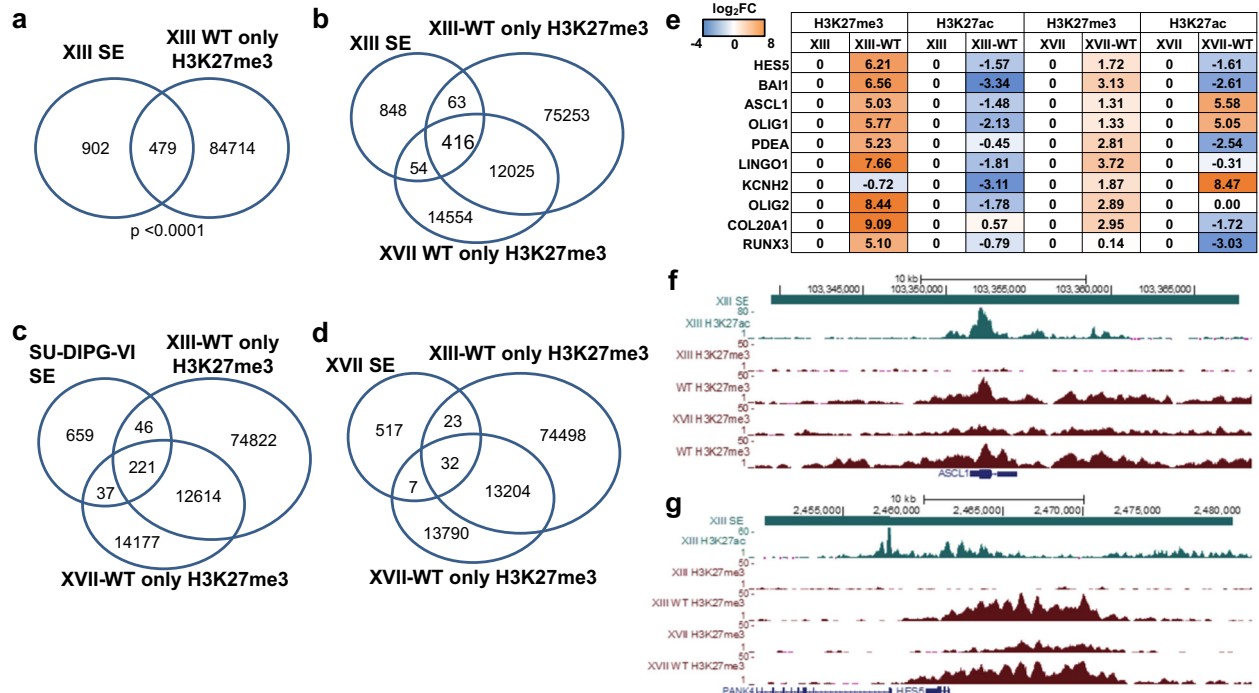

**Fig. 3 H3.3-mutated cells have lower H3K27me3 at super-enhancers, including at neurogenesis and NOTCH pathway genes, compared to their H3.3WT isogenic counterparts. a** Overlap of XIII super-enhancers (SE) with H3K27me3 peaks found only in XIII-WT. **b** Three-way overlap of XIII super-enhancers (SE) with H3K27me3 peaks specific to XIII-WT and XVII-WT. **c** Three-way overlap of line VI super-enhancers with H3K27me3 peaks specific to XIII-WT and XVII-WT. **d** Three-way overlap of XVII super-enhancers with H3K27me3 peaks specific to XIII-WT and XVII-WT. **e** H3K27me3 and H3K27ac levels at super-enhancer regions associated with select genes validated by ChIP-qPCR. Ct values were normalized to parental glioma lines, and $\log_2$ fold change over parental is shown. $n = 2$ or 3 biologically independent samples for XIII and XVII pairs, respectively. **f, g** ChIP-seq read pileup for H3K27ac and H3K27me3 in *ASCL1*-associated and *HES5*-associated super-enhancer region for gene-edited glioma lines. One of the two similar replicates is shown per cell line.

thoroughly compared their expression profiles finding that more than half of the genes that exhibit differential expression in G34R cells also undergo similar changes in K27M cells when compared with their own isogenic cell lines (Fig. 4g, h). Comparing across all variations of our HA and SF188 cells, we identified 147 genes that are commonly overexpressed in all H3.3 mutant (K27M and G34R) HA and SF188 cell lines (Fig. 4g), but none were commonly downregulated across all mutant lines (Fig. 4h).

In addition to the 147 commonly overexpressed genes across G34R and K27M HA and SF188 lines, there are 239 genes that are overexpressed only in HA-G34R and SF-G34R. GO analysis of these 386 G34R-associated genes indicates that G34R lines have some distinct gene clusters not commonly overexpressed in K27M lines (Fig. 4f, i), including elevated cell adhesion, cell membrane, phosphoprotein, and glycoprotein clusters, while synapse, cell junction, and ion channel gene clusters were elevated only in K27M lines. Thus our data also point to some potential unique mechanisms of G34R and K27M mutations, despite the conserved impact on neurogenesis and NOTCH signaling genes.

**Histone marks are altered in NOTCH and neurogenesis genes.** To validate our ChIP-seq results and define how the changes they revealed relate to other histone marks, we performed ChIP coupled with qPCR (qChIP) for H3.3, H3K27me3, H3K27ac, and H3K36me3 at our selected neurogenesis and NOTCH signaling genes in our full panel of cells (Supplementary Fig. 6c–f). H3.3 was significantly more enriched at the promoters of almost all of these genes probed in HA-K27M/G34R and SF-K27M/G34R cells versus otherwise isogenic WT parental lines. A reciprocal pattern was observed as H3.3 reverted-to-WT gliomas exhibited almost

uniformly decreased H3.3 in the gene promoters (Supplementary Fig. 6c). H3K27me3 was strongly reduced at the promoters in mutant versus WT cells, whereas XIII-WT and XVII-WT cells exhibited increased H3K27me3 (Supplementary Fig. 6d). While H3K36me3 and H3K27ac patterns were relatively less consistent, we observed a general trend of reduced H3K36me3 and H3K27ac at the promoters of mutant H3.3 target genes in H3.3 mutant cells (Supplementary Fig. 6e, f). These H3.3 mutant-specific trends in histone mark levels at promoters of NOTCH pathway and neurogenesis genes corresponded with increased NOTCH1 and ASCL1 protein (as well as RNA) levels across mutant H3.3 cells compared to WT cells, more so for K27M than for G34R (Supplementary Fig. 4e, Supplementary Table 1, Supplementary Fig. 8).

**NOTCH inhibition exhibits selectivity for H3.3 mutant cells.** To determine whether the oncogenic properties specifically of H3.3 mutant cells could be attenuated by targeting the overlapping NOTCH and neurogenesis pathways, we treated our panel of gene-edited and parental cells with DAPT, a γ-secretase inhibitor that inhibits NOTCH1 cleavage (Fig. 5). We found that XIII and XVII glioma cells were more sensitive to DAPT treatment than their WT-reverted isogenic counterparts, while pairs of HA and SF188 H3.3WT and mutant cell lines each exhibited little shift in DAPT sensitivity (Supplementary Table 6). When combined with irradiation treatment that mimics some clinical treatments of gliomas (2 GY of radiation immediately prior to DAPT treatment), we found that HA-K27M, SF-G34R, XIII, and XVII all exhibit significantly lower IC50s compared to their H3.3WT counterpart cells (Fig. 5a, Supplementary Table 6). In

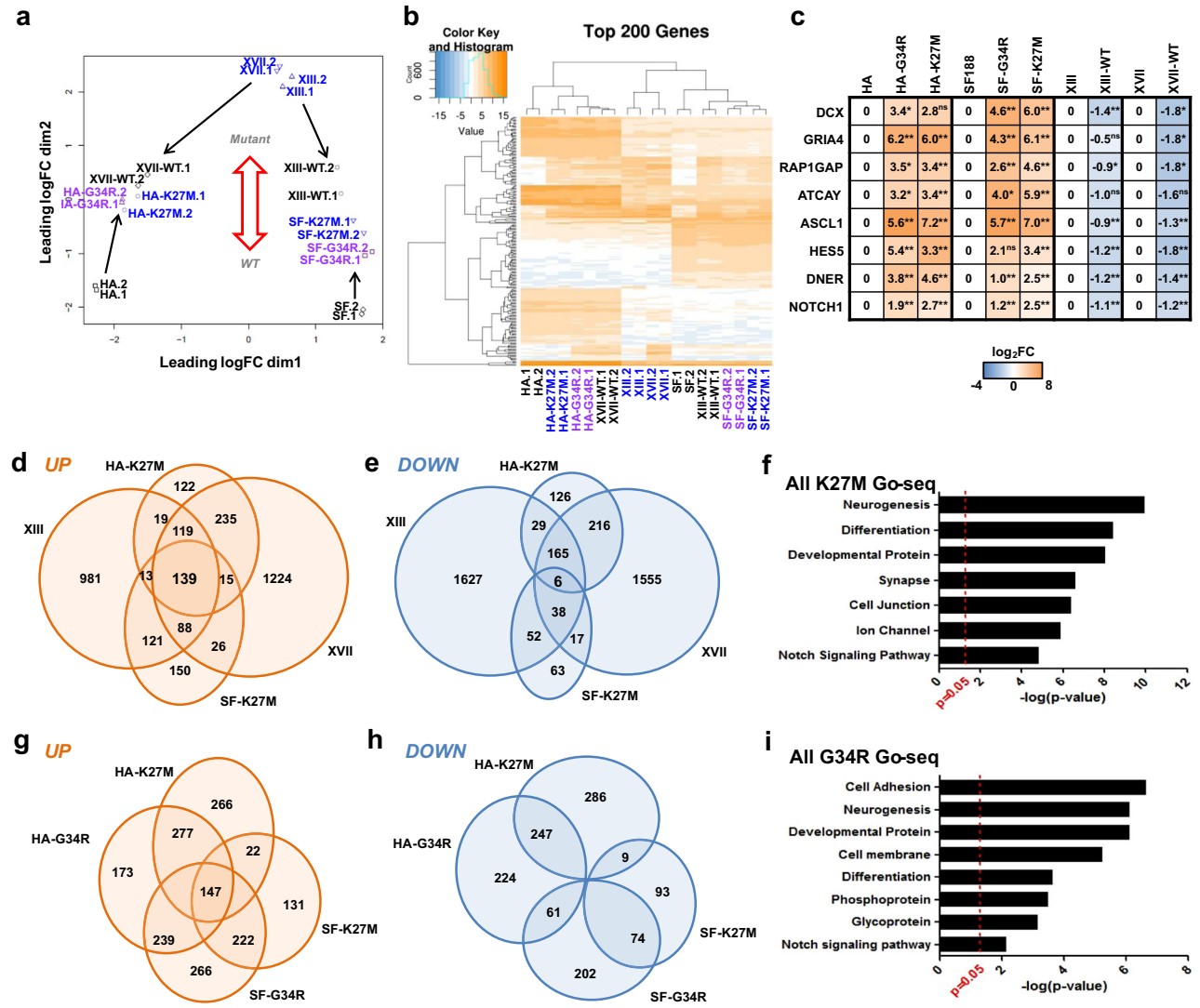

**Fig. 4 Transcriptomic analysis implicates neurogenesis and other pathways in mutant H3.3 gliomagenesis. a** Multi-dimensional scaling plot of cell panel gene expression. Biological replicate samples are denoted by .1 and .2 suffixes; label colors reflect H3.3 mutation status: WT (black), G34R (purple), K27M (blue). **b** Hierarchal heatmap of top 200 differentially expressed genes. **c** Heatmap of RT-qPCR expression values performed on select neurogenesis and NOTCH pathway genes across our cell panel. Relative $\log_2$FC values normalized to respective parental cells are tabulated. Data are the means of $n = 3$ biologically independent samples; *$p < 0.05$, **$p < 0.01$, ns = not significant. **d, e** Venn diagrams of overlapping genes that are significantly upregulated and downregulated in H3.3K27M mutant cells compared to H3.3WT counterparts, respectively. **f** Top gene ontology clusters that are enriched across all H3.3K27M cell lines relative to their matched WT cells. **g, h** Venn diagrams comparing genes that are significantly upregulated and downregulated in H3.3G34R and H3.3K27M cells. **i** Top gene ontology clusters that are enriched across both H3.3G34R cell lines relative to their matched WT cells with $p$ values at right.

contrast to DAPT, treatment with histone deacetylase inhibitor Panobinostat largely killed cells regardless of H3.3 mutation status (Fig. 5b, Supplementary Table 6). These findings show that, in comparing otherwise isogenic cell lines, DAPT treatment exhibits selectivity for H3.3 mutant cells, which have elevated NOTCH signaling and ASCL1 levels.

**NOTCH pathway intervention modulates glioma viability.** To test whether ASCL1 is acting downstream of K27M, we performed small interfering RNA (siRNA) knockdown of ASCL1 in our parental and gene-edited glioma lines (Fig. 5c). ASCL1 knockdown caused large, statistically significant reductions in cell viability in XIII and XVII parental gliomas, whereas XIII-WT and XVII-WT exhibited smaller reductions (Fig. 5d). Thus cellular sensitivity to reduced ASCL1 tracked with K27M mutation status

and with the elevated expression of ASCL1 in K27M-bearing cell lines (Figs. 4c and 5d, Supplementary Fig. 4e).

We also tested whether exogenous ASCL1 expression could rescue oncohistone phenotypes in reverted-to-WT cells (Fig. 5e). This increased ASCL1 expression resulted in significantly increased cell viability in XIII-WT and XVII-WT (2.5- to 3.5-fold, respectively) suggesting that ASCL1 overexpression mimics the effect of H3.3K27M in the mutant cell lines in inducing NOTCH signaling and thereby increases cell cycling and viability (Fig. 5f).

In addition, we performed siRNA knockdown of RBPJ, a transcription factor downstream of NOTCH, for which a single siRNA was sufficient to achieve >60% knockdown in XIII and XVII cell lines but was less effective in XIII-WT and XVII-WT (Fig. 5g). This level of RBPJ knockdown led to significantly reduced viability in both XIII and XVII (Fig. 5h).

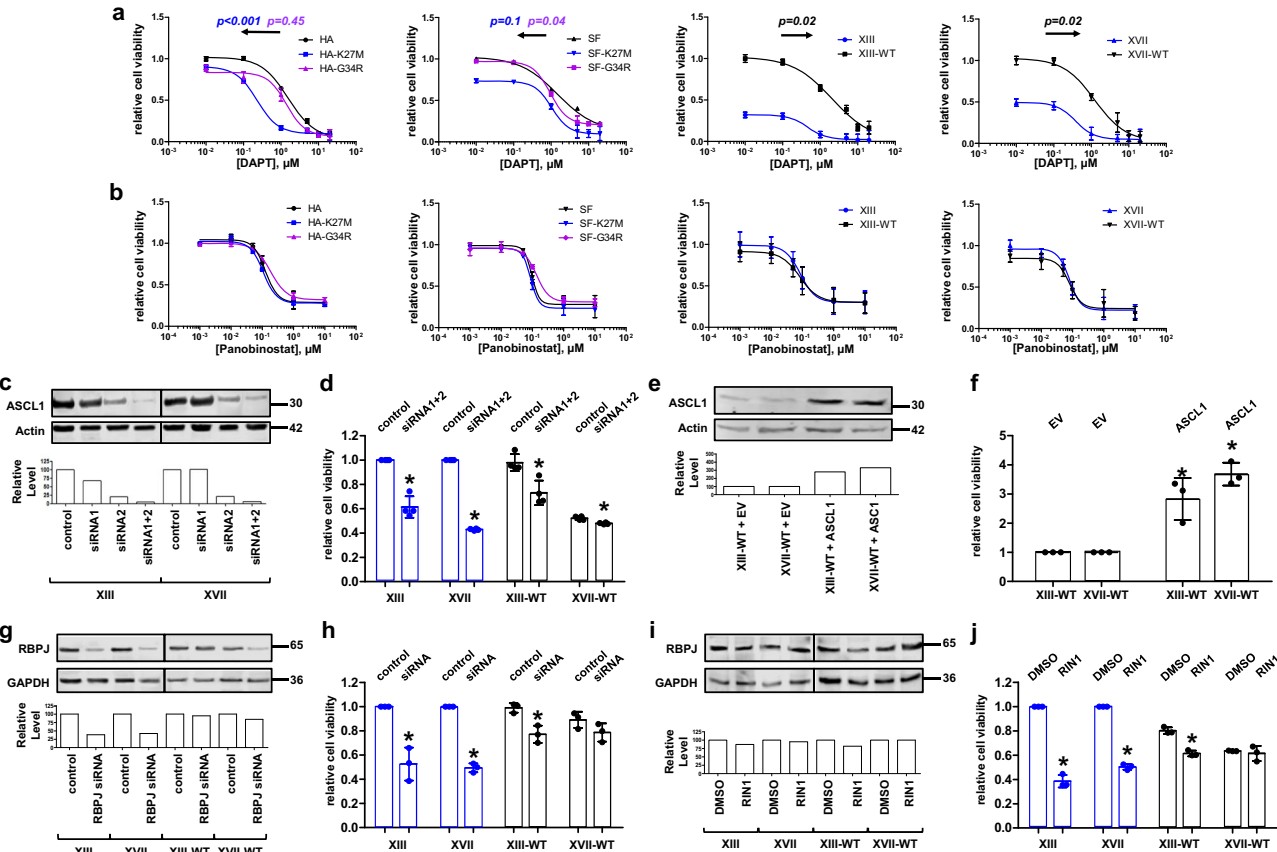

**Fig. 5 siRNA knockdown and drug inhibition of NOTCH signaling pathway reduces viability of H3.3-mutated glioma cells. a** Dose–response curves of DAPT treatment on our cell panel. Curves were fitted using the "log(inhibitor) versus response–variable slope" model and IC50s were calculated using the GraphPad Prism software. Arrows denote the direction of IC50 shift upon gene editing where applicable. **b** Dose–response curves of Panobinostat treatment on our cell panel. For **a**, **b**, p values comparing LogIC50s were calculated by unpaired t test using GraphPad QuickCalcs with $n = 3$ biologically independent samples. **c** Western blot of ASCL1 knockdown in subgroups of our panel of cells using siRNA for ASCL1, with quantification of ASCL1 levels exhibiting strong knockdown in XIII and XVII lines. siRNA targeting noncoding RNA from the same vendor as our targeting siRNA (Thermo) was used as control. **d** Cell viability measured using CellTiterGlo assay (Promega) of cells under siRNA knockdown of ASCL1 plotted as dot plot with mean ± sd; $n = 4$ biologically independent samples. **e** Western blot of ASCL1 levels with transient transfection of empty vector or pCS2-HA-ASCL1 plasmid leading to overexpression of ASCL1 in XIII-WT and XVII-WT cells. **f** Cell viability of XIII-WT and XVII-WT cells with overexpression of ASCL1, mean ± sd; $n = 3$ biologically independent samples. **g** Western blot of RBPJ knockdown in our panel of cell lines. RBPJ knockdown was the strongest in XIII and XVII cells, which exhibit higher RBPJ expression than the H3.3WT-reverted cells. **h** Cell viability of glioma cell lines under RBPJ knockdown, mean ± sd; $n = 3$ biologically independent samples. **i** Western blot of RBPJ levels across cell panel under RIN1 (RBPJ inhibitor 1) treatment (2 μM, 72 h). RIN1 does not significantly impact RBPJ protein levels. **j** Cell viability of glioma cells under RIN1 treatment, mean ± sd; $n = 3$ biologically independent samples. All p values were calculated by Student's t test comparison of treated cells versus control or vehicle. *$p < 0.05$.

Recently, a study reported the development of a small molecule inhibitor (RIN1) against RBPJ[24]. We performed RIN1 treatment of our glioma cells, and while RIN1 did not have a noticeable effect on RBPJ expression (Fig. 5i), RIN1-treated XIII, XIII-WT, and XVII cells exhibited significantly reduced cell viability (Fig. 5j). Taken together, these assays show that knocking down or inhibiting different components of the NOTCH pathway (ASCL1, NOTCH cleavage, RBPJ) reduces viability of our cell lines, particularly the H3.3K27M mutant lines.

**Differential tumor development in mouse brain xenografts.** We performed orthotopic xenograft assays in mice using our panel of edited cells injected into the pons. Brain tissue was collected for immunofluorescence (IF) staining and histological analysis. IF staining for human cells in the mouse brains using an anti-human nuclei antibody confirmed the engraftment of human cells in the mice (Supplementary Fig. 7). In addition, by staining the tissue with an H3.3K27M-specific antibody, we confirmed that mutant H3.3 protein was expressed only in the gene-edited

HA-K27M and SF-K27M lines as well as the parental XIII and XVII gliomas (Supplementary Fig. 7c, f, g, i).

Mice injected with HA cells did not develop tumors (Fig. 6a, Supplementary Figs. 7). While a few tumor-like cells (clusters of enlarged cells in the leptomeninges) were found in a few HA-K27M xenografted mice, the number of tumors, tumor size, and extent of tumor infiltration were not statistically different than in HA-injected mice, reiterating previous findings that H3.3K27M alone is not clearly tumorigenic[15] (Fig. 6a, b, Supplementary Data 3). Notably, HA-G34R xenografted mice developed diffuse small tumors that were significantly higher in number and atypia severity than HA xenografted mice (Fig. 6a, b, Supplementary Data 3), suggesting that G34R may be somewhat more tumorigenic on its own in this model. Despite the presence of these small tumors, all HA, HA-K27M, and HA-G34R xenografted mice survived to the maximum 6-month endpoint without symptoms (Fig. 6c).

In contrast, SF-K27M-injected mice developed invasive, solid tumors and at a significantly higher rate than parental SF188-

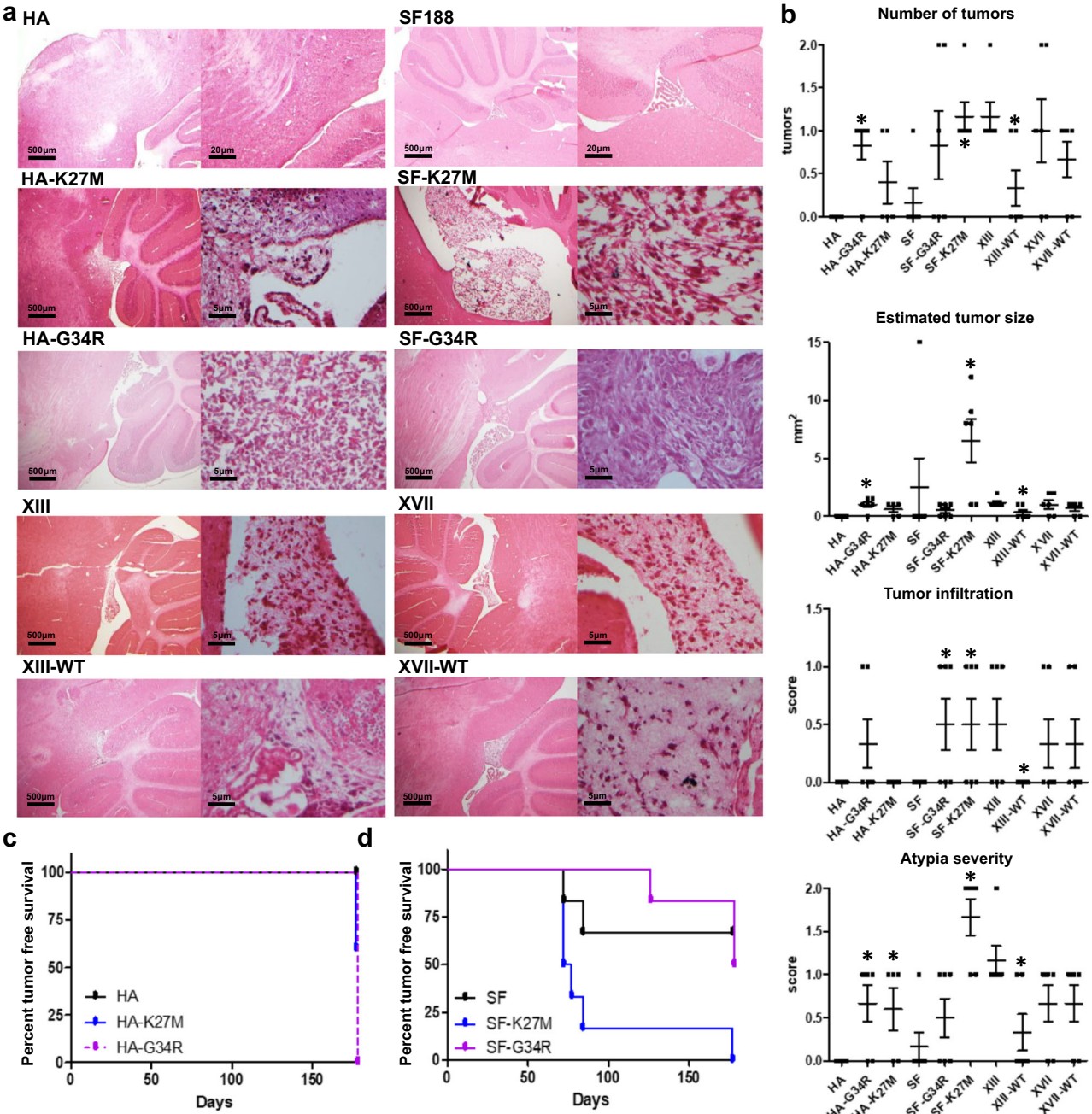

**Fig. 6 H3.3K27M and G34R glioma cells engraft in mouse brain and develop tumors at a higher rate than isogenic H3.3WT counterparts. a** Hematoxylin and eosin-stained images of mouse brains xenografted with our cell panel. All cell lines engrafted to at least some minimal detectable extent, but no clearly identifiable tumor cells were observed in HA samples and a tumor was present in only one of the six SF188 samples. Various clusters of tumor cells were found in the brains with all other cell lines but to a greater degree and with greater disease severity in SF-K27M and the parental XIII and XVII glioma lines. **b** Histological scoring of the number of tumors, estimated tumor size, tumor infiltration, and atypia severity across xenograft samples plotted as mean ± sd of scores; $n = 6$ mice per cell line with *$p < 0.05$ denoted for comparison of parental to gene-edited cell lines, Student's $t$ test. Scoring of individual tissue samples are in Supplementary Table 8. **c**, **d** Tumor-free survival curves of xenograft mice; $n = 6$ mice per cell line, *$p < 0.05$, log-rank test. Note: Survival curves for mice xenografted with other cell lines were not plotted as all mice survived to the maximum study duration (like the plot for HA-xenografted mice in **c**) and exhibited no behavioral signs of tumor despite some mice developing tumor as evidenced by the postmortem IF staining and histology.

injected mice (five out of six with solid tumors versus one out of five, respectively) (Fig. 6a, b, Supplementary Data 3). The tumors were mostly anterior to the cerebellum and often merged with or invaded into the pons and cerebellar tissue, sometimes severely disrupting the cerebellum (Fig. 6a, Supplementary Fig. 7). SF-K27M-injected mice exhibited significantly reduced tumor-free survival compared to SF188-injected mice (Fig. 6d). SF-G34R-

injected mice only developed small, diffuse tumors that were nonetheless histologically observable and exhibited more severe tumor infiltration than parental SF188 xenograft samples (Fig. 6a, b, Supplementary Data 3).

Mice injected with XIII, XIII-WT, XVII, and XVII-WT exhibited no behavioral signs of tumor development and survived to the 6-month endpoint of the study, but histological analysis

indicated that a higher fraction of the XIII and XVII xenografted mice developed clusters of tumors as compared to their XIII-WT and XVII-WT counterparts (Fig. 6a, b, Supplementary Data 3). These H3.3 mutant tumors were found to have the typical morphology of high-grade glioma, and gene-editing to WT generally reduced the severity of the phenotype (Fig. 6a, b, Supplementary Fig. 7). XIII-WT in particular exhibited significantly reduced number of tumors, reduced tumor size, less atypia, and lowered infiltration compared to XIII when xenografted into mice brain (Fig. 6b, Supplementary Data 3). XVII-WT cells also followed the trend of reduced tumorigenesis potential but to a lesser degree than XIII-WT.

## Discussion

To further uncover mutant H3.3 gliomagenesis pathways, we used CRISPR-Cas9 to convert H3.3WT astrocytes and glioma cells to K27M and G34R while in parallel reverting the K27M mutation to WT in two pediatric DIPG cells. Our study identified potentially new (to our knowledge) mechanisms but also recapitulated several of the previously identified properties of mutant H3.3 pediatric gliomas. The latter include decreased global H3K27me3[8,15,25] and increased *MYCN* (notably in both our K27M and G34R-mutated cells) that was also observed in studies of pediatric GBM specifically with H3.3G34(R/V) mutation[13,26] but had not been previously reported in K27M cells. Though we do not observe a decrease in cell proliferation in XIII-WT, our XVII-WT reverted cell line exhibited markedly decreased cell proliferation and morphological changes in culture, suggesting that the oncogenic effects of K27M are sometimes reversible, an encouraging finding at the clinical oncology level. This difference in effect on cell growth from gene editing H3.3 in cell lines from different patients could point to additional genetic differences that are cell line specific that could modulate the exact phenotypic outcome. However, the large-scale epigenetic and transcriptomic alterations due to mutation of H3.3 in our study are consistent across all pairs of isogenic cell lines.

The trend of increased H3.3 and total H3 levels in many of our H3.3-mutated lines was striking. These were confirmed by both Western blot and qPCR of *H3F3A* and *H3F3B*, where *H3F3B* expression was elevated in some cases in lines with higher H3.3 protein levels. Because changes in global histone levels could somewhat impact interpretation of histone marks changes, all histone mark protein level interrogations were normalized to total H3 or H3.3 levels accordingly to try to account for differences in endogenous histone H3 family expression. In addition, we performed H3.3 ChIP-seq alongside our H3K27me3 ChIP-seq, finding that the largest differences in K27me3 binding between isogenic gene-edited mutant and WT lines occur independently of H3.3 binding and thereby should not be confounded by changes in H3.3 levels.

ChIP-seq analysis along with ChIP-RT-qPCR validation of our H3.3 mutant and reverted-to-WT glioma cells revealed striking overlap of regions that are depleted of H3K27me3 in H3.3 mutant cells with previously identified super-enhancer regions in DIPGs[17]. Critically, several of these regions are associated with NOTCH pathway genes and were overexpressed in H3.3 mutant cells. Reversion of H3.3K27M to WT in these gliomas alone is sufficient to restore H3K27me3 levels and reduce gene expression at these super-enhancer regions. Our transcriptomic data suggest that H3.3 mutations primarily drive tumorigenesis via overexpression of oncogenic pathways rather than due to loss of tumor-suppressor functions. Taken together with global and gene-specific decreases of H3K27me3 marks, loss of gene repression at specific cell signaling and neurogenesis pathway genes is potentially a key mechanism of mutant H3.3

tumorigenesis (Fig. 7). The specific factors most consistently associated with mutant H3.3 included ASCL1 and NOTCH1, notably with the induction of ASCL1 protein by mutant H3.3 being at least partially NOTCH pathway dependent. More generally, the function of ASCL1 is highly context dependent and at times promotes precursor differentiation or proliferation[27-29]. The elevated ASCL1 in mutant H3.3 glioma may be accompanied by selective inhibition of its differentiation-promoting, pro-neural function. Taken together, these findings support a model wherein mutation of H3.3 leads to loss of H3K27me3 at super-enhancer regions, including at NOTCH and neurogenesis pathway genes, leading to their derepression and aberrant overexpression, which ultimately contributes to gliomagenesis through a proposed aberrant developmental pathway mechanism (Fig. 7).

GO analysis identified some unexpected categories of genes, such as sensory perception of smell and keratinization that are enriched for H3K27me3 by ChIP-seq in H3.3 mutant cells and these genes are often clustered along the chromosome. For example, the enriched keratinization and peptide crosslinking categories are primarily composed of genes from the epidermal differentiation complex, expressed during epidermal differentiation[30]. These clusters often have high levels of H3K27me3 when the member genes are not being expressed[31], which suggests a potential role for WT H3.3 in the regulation of expression from gene clusters during normal development. In addition, we observed notable changes in cell adhesion in culture corresponding with changes in expression of cell junction and cell adhesion genes in our gene-edited lines. While these changes could point to large-scale cellular remodeling processes such as differentiation machinery playing a role, it could very well be a downstream effect of H3.3 mutation, and it would be difficult for one to determine the cause and effect—whether adhesion and larger-scale morphological changes or genomic/epigenomic changes occur first.

A recent study performed CRISPR-Cas9 knockout of mutant *H3F3A* in two H3.3K27M glioma lines and overexpressed recombinant H3.3K27M[9]. Overall, comparing our data to theirs (Supplementary Fig. 5c) indicate that the two modes of modeling H3.3K27M mutations via knockout of the entire mutant allele (their approach) or point mutation reversion to WT (our approach) resulted in some different epigenetic outcomes, perhaps in part due to the different relative levels of WT and mutant *H3F3A* alleles. In addition, their introduction of mutant H3.3 via exogenous overexpression also resulted in H3K27me3 changes distinct from what we observed by targeted point mutation.

The rarity of H3.3G34R/V mutations in patients has made them less well studied, though G34 mutation may affect the H3K36me3 mark either globally and/or at specific gene loci including *MYCN*[32]. In our study, total levels of H3K36me3 exhibited only variable changes with H3.3 mutation. Overall, our data support a new (to our knowledge) model in which K27M and G34R/V mutations contribute to tumorigenesis via at least partially overlapping mechanisms, potentially through G34R/V affecting modification of the nearby K27 residue in *cis*. The major altered, downstream genetic pathways are shared between K27M and G34R gene-edited cells, including neurogenesis and NOTCH signaling genes. Furthermore, patterns of loss of H3K27me3 (and to a lesser extent the other histone marks analyzed) at these target gene loci in HA-G34R and SF-G34R cell lines are strikingly similar to that observed in respective K27M-mutated lines. Nonetheless, numerous other differentially expressed genes unique to our engineered G34R cell lines were evident.

Despite some recent advances in mutant H3.3 therapy research, pediatric glioma patient outcomes remain devastatingly poor due to limited treatment options. We found that DAPT and RIN1 exhibit selectivity for inhibiting growth of H3.3 mutant cells

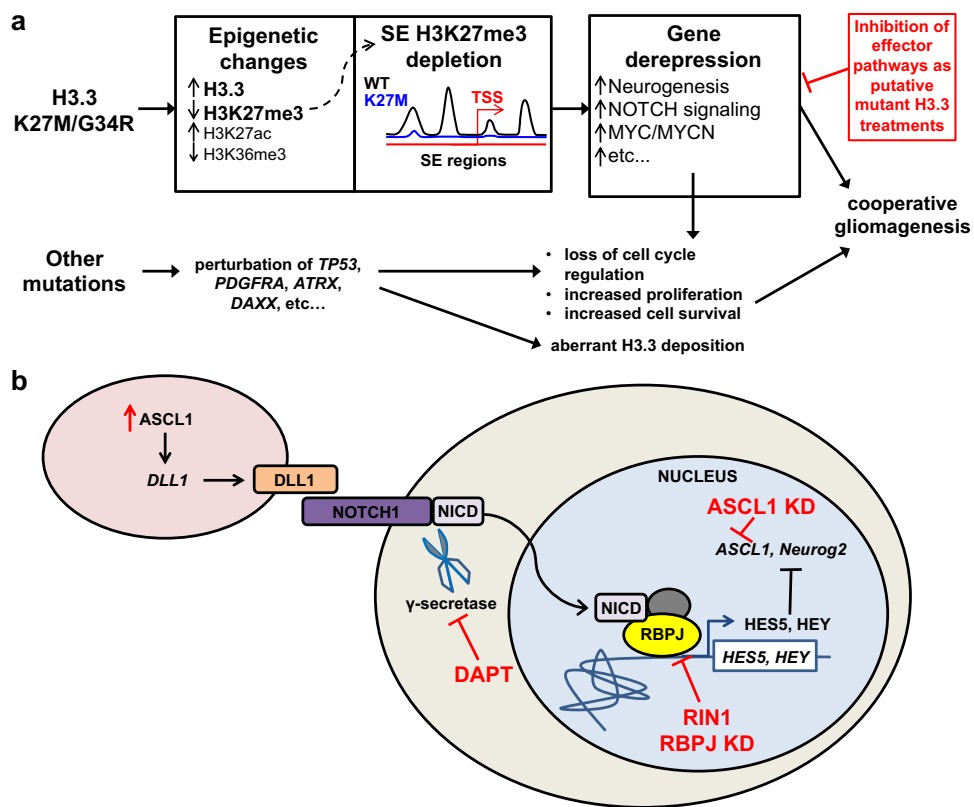

**Fig. 7 A model of pediatric gliomagenesis via cooperative H3.3 mutation with additional oncogenic perturbations leading to runaway NOTCH and neurogenesis signaling. a** Differential H3K27me3 binding events to super-enhancer (SE) regions are depicted as a cartoon model leading to derepression of downstream pathways and feed-forward overexpression of oncogenic NOTCH signaling and neurogenesis pathways. **b** NOTCH signaling pathway consists of upstream ASCL1 expression, leading to inter-cell activation and cleavage of NOTCH1 by γ-secretase, with eventual translocation of NICD to the nucleus where it interacts with transcription factors, including RBPJ, thereby inducing transcription activation of genes, such as *HES5*. Increased NOTCH signaling has been described as a hallmark of some cancers leading to increased cell cycle transitions. We showed H3.3 mutant glioma cells have elevated expression of NOTCH pathway genes. Transient overexpression (OE) of ASCL1 is sufficient to increase cell viability of H3.3WT-reverted gliomas. siRNA knockdown (KD) of ASCL1 and RBPJ as well as drug inhibition of γ-secretase with DAPT or RBPJ with RIN1 lead to decreased cell viability of H3.3K27M glioma cells in culture. While this model shows ASCL1 OE upstream and ASCL1 KD downstream in the pathway, changes in ASCL1 expression could act in either position to affect cell viability.

over their paired H3.3WT counterparts. Data from our isogenic xenograft system support the current model that mutant H3.3 is not sufficient to drive tumor formation but can confer more aggressive cancer phenotypes. A caveat to keep in mind is that, while we injected our gene-edited cells into the right hemisphere pons, DIPGs and GBMs typically arise in different regions of the brain as do K27M and G34R-bearing tumors, which are typically midline and hemispheric tumors, respectively. We chose to keep the same injection coordinate for consistent comparison across our panel of cell lines, but further exploration of intracranial xenograft with these cells could yield insightful knowledge about the sub-cranial localization of tumor development of these various gliomas. In addition, the panel of isogenic cells may also be a useful tool for drug evaluation both in vitro and in vivo. Further screening or development of drugs targeting the pathways identified in this and other studies, especially in combination with current promising investigational treatments for pediatric glioma including immunotherapy[33,34], may yield a more effective therapeutic arsenal.

## Methods

**Cell culture and characterization**. Patient-derived pediatric glioma cells were generously provided by Dr. Michelle Monje (SU-DIPG-XIII and SU-DIPG-XVII) and Dr. C. David Allis (SF188) (Supplementary Table 1). SF188 was cultured in Dulbecco's modified Eagle's medium (DMEM, Lonza) supplemented with 10%

fetal bovine serum (FBS; Hyclone) and 1% GlutaMax (Invitrogen). All DIPG cells were cultured in Tumor Stem Medium[17], which contains DMEM/F12 1:1 (Invitrogen), Neurobasal-A (Invitrogen), 10 mM HEPES (Invitrogen), 1× MEM sodium pyruvate (Invitrogen), 1× MEM nonessential amino acids (Invitrogen), 1% GlutaMax (Invitrogen), human basic fibroblast growth factor (20 ng/mL) (Shenandoah), human epidermal growth factor (20 ng/mL) (Shenandoah), human platelet-derived growth factor (PDGF)-A and PDGF-B (20 ng/mL) (Shenandoah), heparin (10 ng/mL) (StemCell Technologies), and B27 without Vitamin A (Invitrogen). Human astrocyte brainstem control cell line (HA) was purchased from ScienCell (Cat #1840) and cultured in astrocyte medium (ScienCell Cat #1801) supplemented with astrocyte growth supplement (ScienCell Cat #1852) and 2% FBS (ScienCell Cat #0010). All cells were routinely cultured in media containing a portion of conditioned medium (mixing half conditioned and half fresh media upon each passage) and dissociated with TrypLE Express (Invitrogen) supplemented with DNase I (10 μg/mL) (Stem Cell Technologies). HA and SF188 are adherent and grow as monolayer, whereas the DIPG lines were generally grown in suspension flasks as tumorspheres, except when they underwent transfection, for which they were dissociated and plated on laminin-coated plates as described subsequently. We confirmed the *H3F3A* status (mutant or WT) of the starting parental cells in our panel via Sanger sequencing of genomic DNA (Supplementary Table 1) and performed a variety of characterizations of the panel. In addition to phenotypic cell growth observations, we performed qPCR of the two histone H3.3 genes (*H3F3A*, *H3F3B*) and the two dominant *MYC* family genes expressed in the nervous system, which are linked to neuronal tumors (*MYC*, *MYCN*)[35], and western blots for H3.3, MYC, and MYCN (Supplementary Table 1).

**CRISPR-Cas9 gene editing**. Guide sequences were designed using an online CRISPR design tool (crispr.mit.edu). In each case, a 200-nucleotide sequence centered on K27 or G34 was uploaded to the server, which finds PAM sites and

generates a ranked list of all guides in the query sequence in order of score determined by on-target activity minus a weighted sum of predicted off-target hits in the genome. Using the results generated for Cas9-nickase (Cas9n), which requires two guides targeting two Cas9n molecules to adjacent regions in order to mimic a double-strand break to stimulate repair and editing, we selected the top three pairs of guides near our desired sites of gene edits. The selected guide pairs targeting *H3F3A* were cloned into pSpCas9n(BB)-2A-GFP (PX461) and pSpCas9n(BB)-2A-Puro (PX462) plasmids[36], which in each case delivers Cas9n and guides in the same plasmid upon transfection into mammalian cells. With one guide cloned into the green fluorescent protein (GFP)-containing plasmid and one in the puromycin resistance conferring plasmid, we were able to select and screen for cells that are both puromycin resistant and exhibit green fluorescence after transfection.

Guide sequences used:
Guide #1: 5'-GAGTGCGCCCTCTACTGGAG-3'
Guide #2: 5'-CTTCCTGGGTGCTTTACCAC-3'.
Homology directed repair (HDR) templates used:
*H3F3A*-WT:
GCATATGGTGATTTTTGATTTTTCAATGCTGGTAGGTAAGTAAGGAG
GTCTCTGTACGATGGCACGAACAAAGCAGACTGCCCGCAAATCGACAGG
AGGTAAAGCACCGAGAAAGCAACTCGCTACAAAAGCCGCTCGC**AAG**AG
TGCGCCCTCTACAGGCGGAGTGAAGAAACCTCATCGTTACAGGTATTAA
AAAACAGGAAAAAAATGGGACAAAGTCTCTCTTGTATG.

*H3F3A*-K27M:
GCATATGGTGATTTTTGATTTTTCAATGCTGGTAGGTAAGTAAGGAG
GTCTCTGTACGATGGCACGAACAAAGCAGACTGCCCGCAAATCGACAGG
AGGTAAAGCACCGAGAAAGCAACTCGCTACAAAAGCCGCTCGC**ATG**AG
TGCGCCCTCTACAGGCGGAGTGAAGAAACCTCATCGTTACAGGTATTAA
AAAACAGGAAAAAAATGGGACAAAGTCTCTCTTGTATG.

*H3F3A*-G34R:
GCATATGGTGATTTTTGATTTTTCAATGCTGGTAGGTAAGTAAGGAG
GTCTCTGTACGATGGCACGAACAAAGCAGACTGCCCGCAAATCGACAGG
AGGTAAAGCACCGAGAAAGCAACTCGCTACAAAAGCCGCTCGCAAGAG
TGCGCCCTCTACAGGC**AGA**GTGAAGAAACCTCATCGTTACAGGTATTAA
AAAACAGGAAAAAAATGGGACAAAGTCTCTCTTGTATG.

HDR after targeted cutting of DNA by Cas9n was facilitated by using single-strand DNA oligos (ssODNs) containing the desired mutations as homologous repair templates. Single-stranded donor oligonucleotide (ssODN) templates were designed with the desired mutations to facilitate HDR after cleavage of *H3F3A* by Cas9n. All ssODNs span the same sequence (140 nt upstream and 60 nt downstream of the codon encoding K27) and only differ by the mutations they contain and are designed to induce (*H3F3A*-K27M, G34R, or WT). Starting with a plasmid containing the HDR template sequence (pcDNA3.1(+) with *H3F3A* fragment cloned in, pcDNA-HDR), the desired K27M, G34R, or WT HDR templates were generated via QuikChange site-directed mutagenesis PCR. ssODNs were then made in-house using a dual-PCR strategy, whereby corresponding double-stranded oligonucleotides were first PCR amplified from the pcDNA-HDR plasmids. After running the product on agarose gel to purify the band corresponding to the double-stranded product, a second asymmetrical PCR[37,38] was run with only one primer to generate ssODNs that were column purified prior to use for transfection.

Two days prior to transfection, 6-well plates were laminin coated (10 µg/mL) at 4 °C overnight. One day prior to transfection, laminin was removed and $1 \times 10^6$ cells were plated into each well. The next day, cells were co-transfected with a combination of pSpCas9n(BB)-2A-GFP-guideA (0.5 µg/well), pSpCas9n(BB)-2A-Puro-guideB (0.5 µg/well), and ssODN HDR (1 µg/well) template using XtremeGene-HP (Sigma). Cells transfected with Cas9n-encoding plasmid but no targeting guides and no ssODN were processed in parallel for each cell line and used as controls for all analytical comparisons (Supplementary Fig. 2a).

**Generation of near-clonal gene-edited cell lines**. Cells transfected with CRISPR-Cas9 plasmids, guides, and HDR templates were screened for evidence of GFP expression 24 h post-transfection and underwent puromycin selection (0.3 µg/mL) for 72 h thereafter (Supplementary Fig. 2a). The surviving cells were then expanded until confluent in 6-well plates and subsequently dissociated and replated in 96-well plates in a limiting-dilution manner (Supplementary Fig. 2b). The minimal plating number of cells was optimized to be ~20 cells per 96-well plate well (at a density of <20 cells per well, the cells generally did not grow and expand). Upon further growth, cells were expanded from 96- into 24-well plates. Upon achieving near confluency in 24-well plates, a portion of each of subcloned well was harvested to extract genomic DNA, from which the mutation-containing region of *H3F3A* was PCR amplified and cloned into plasmids using a Topo-TA Cloning Kit according to the manufacturer's instructions. These plasmids were transformed into DH5α bacteria that were then plated onto LB agar containing ampicillin. Single colonies were miniprepped for plasmid DNA and sequenced. A minimum of ten colonies were sequenced for each subclone and analyzed as representative of the *H3F3A* mutant status in individual subclones. In each case, the subclone with the highest percentage of the desired gene-edited alleles was then expanded and replated into 96-well plates for iterative limiting-dilution subcloning. This process was repeated until a 100% gene-edited subclone was achieved for each gene edit

intervention based on sequencing at least 10 bacterial colonies from Topo cloning (Supplementary Fig. 2c, d), yielding clonal or near-clonal lines.

**Screening for on-target indels and off-target gene editing**. Hypothetically, inadvertent insertions/deletions (indels) that arise maybe be missed when Topo-TA cloning and screening smaller amplicons for the desired point mutations if the indels occur outside the narrow amplified region or disrupt the binding of PCR primers. To address this possibility, we PCR amplified a 2-kb region spanning upstream and downstream of the targeted cut site of H3F3A and detected no size shifts in any of our gene-edited cell lines as compared to their respective parental lines upon running on an agarose gel (Supplementary Fig. 3b). In addition, the potential presence of off-target genetic modifications was assessed by isolating genomic DNA from each of our edited cell lines and sequencing PCR amplifications of the top off-targets predicted by an online guide design tool (crispr.mit.edu). There are no predicted potential off-target DNA sequences common to the two guides we used in conjunction with Cas9 nickase (which requires coordinate binding of two proximal guides to mimic a double-strand cut and induce DNA repair leading to gene editing) (Supplementary Fig. 3c). Nonetheless, to check the potential for unexpected single-guide-mediated off-target Cas9n activity, we further assessed the only predicted off-target even from a single guide annotated in NCBI corresponding to a transcribed gene (NM_001145553.1). Importantly, this record was recently updated and suppressed owing to lack of evidence for protein product, but regardless we PCR amplified a 499-nucleotide region of the genomic DNA centered on the predicted binding sequence and confirmed via Topo-TA cloning and sequencing that no mutations were induced in this region by our *H3F3A*-targeting guides (Supplementary Fig. 3d).

**Western blotting**. Whole-cell lysates, nuclear extracts, or acid extracts were prepared from cells pelleted after washing with phosphate-buffered saline (PBS). Samples were diluted in NuPAGE LDS Sample Buffer (Novex) with 10% 2-mercaptoethanol. Protein samples were run on NuPAGE Bis-Tris mini gels (Novex), and membrane transfer was performed in 1× MES NuPAGE buffer (Novex). The following antibodies were used for probing of membranes: Primary antibodies–mouse anti-c-Myc (Santa Cruz Biotechnology, #sc-40), mouse anti-n-Myc (Santa Cruz Biotechnology, #sc-53993), mouse anti-H3 (Upstate/Millipore Sigma, #05-499), rabbit anti-H3.3 (Millipore Sigma, #09-838), rabbit anti-H4 (Millipore Sigma, #04-858), rabbit anti-H3K27M (Millipore Sigma, #ABE419), mouse anti-Cas9 (Cell Signaling, #14697), rabbit anti-H3K27me3 (Cell Signaling, #C36B11), rabbit anti-H3.3S31p (AbCam #ab92628), rabbit anti-H3K27ac (Abcam, #ab4729), rabbit anti-K36me3 (Abcam, #ab9050), rabbit anti-NOTCH (Abcam, #ab52627), rabbit anti-ASCL1(MASH1) (Abcam, #ab74065), mouse anti-beta actin (Sigma, #A1978), rabbit anti-RBPJ (Millipore-Sigma #ABE384); secondary antibodies— IRDye 680RD goat anti-mouse (1:10,000; LiCor) and IRDye800CW goat anti-rabbit (1:10,000; LiCor). Blots were imaged, data were captured on a Licor Odyssey CLx, and quantification was performed with the Licor ImageStudio software.

**Reverse transcription qPCR**. RNA was extracted from cells using the NucleoSpin RNA Kit (Macherey-Nagel) from which cDNA was made from each sample using the iScript cDNA Synthesis Kit (Bio-Rad). RT-qPCR was performed using TaqMan Gene Expression Assays (Applied Biosystems; for human *H3F3A*, *H3F3B*, *MYC*, *MYCN*, normalized to *GAPDH*) or Absolute Blue SYBR Green ROX (Thermo-Fisher; for human *ASCL1*, *NOTCH1*, *HES5*, *DNER*, *DCX*, *GRIA4*, *RAP1GAP*, *ATCAY*, normalized to *PPIA*) on a LightCycler 480 (Roche). Primers are listed (5'-to-3') as follows:

| | |
|---|---|
| ASCL1 forward | CAAGCAAGTCAAGCGACAGC |
| ASCL1 reverse | GCCCAGGTTGACCAACTTGAC |
| NOTCH1 forward | CACGCTGACGGAGTACAAGTG |
| NOTCH1 reverse | CAGTGGCAGATGTAGGAGGC |
| HES5 forward | CAGCCCCAAAGAGAAAAACC |
| HES5 reverse | CAGCCATCTCCAGGATGTCG |
| DNER forward | CGAAAACAGGGCAGAAAGTTG |
| DNER reverse | GACTGAGGTGTTCTGTGGCAC |
| DCX forward | CCTTGCTGGCTGCACCTGACGC |
| DCX reverse | CAGTTGGGGATTGACATTCTTGG |
| GRIA4 forward | CTCAAGGAGAGGAAATGCTGG |
| GRIA4 reverse | GAACATTCCCTGTCAGCC |
| RAP1GAP forward | CATTCCATACCCGAGCGTG |
| RAP1GAP reverse | GTGATTTCGTGGTTGGTGCC |
| ATCAY forward | GGAATGGCAACGAACTGGAG |
| ATCAY reverse | GGTGCTCTTGCTCCCCGATG |
| PPIA forward | CTCGAATAAGTTTGACTTGTGT |
| PPIA reverse | CTAGGCATGGGAGGGAACA |

**ChIP and ChIP-seq**. ChIP was performed as described above for qChIP. The following antibodies were used: H3K27me3 (Cell Signaling C36B11) and histone

H3.3 (Millipore Sigma #09-838). After washing, elution, and DNA extraction, the resulting material was used for library preparation for ChIP-seq. Two replicates of H3K27me3 and two replicates of H3.3 ChIP-seq were performed in each of the following cell lines XIII, XIII-WT, XVII, and XVII-WT. An input control was also sequenced for each cell line for normalization. Libraries were prepared with the Nextera Library Prep Kit and sequenced on the NovaSeq with 150 base paired-end sequencing. ChIP-seq reads were aligned to the genome using the Burrows-Wheeler Aligner[39]. MACS2[40] was used to call peaks, with input samples used as the background control and an false discovery rate of 0.05. Differential peak analysis between WT and K27 mutant cell lines was performed with the R package Diff-Bind[41]. Because of a strong outlier sample that needed to be discarded in our H3.3 ChIP-seq, data from lines XIII and XVII were combined for analysis. DAVID was used for GO analysis, and enriched motifs were identified with MEME[42]. ChIP-seq data were uploaded to the Galaxy web platform, and we used the public server at usegalaxy.org for downstream analysis[43]. ChIP-seq data are accessible via Gene Expression Omnibus (GEO; accession series GSE129765).

**ChIP and RT-qPCR.** In performing ChIP, cells were crosslinked with 1% formaldehyde, lysed, and sonicated using a Bioruptor Pico (Diagenode) to generate chromatin fragments <500 bp. For each ChIP, 20–30 μg of sonicated chromatin was used and immunoprecipitated on magnetic Dynabeads (Invitrogen). The following antibodies were used: IgG (Santa Cruz, #sc-2027), mouse anti-H3 (Upstate/Millipore Sigma, #05-499), rabbit anti-H3.3 (Millipore Sigma, #09-838), rabbit anti-H3K27me3 (Cell Signaling, #C36B11), rabbit anti-H3K27ac (Abcam, #ab4729), and rabbit anti-K36me3 (Abcam, #ab9050). Chromatin immunoprecipitated samples were then used as input for qPCR to analyze for enrichment at specific target genes.

For spike-in normalized ChIP-qPCR, 5 μg of sonicated Drosophila chromatin was added to each ChIP sample prior to immunoprecipitation. qPCR enrichment was normalized to Drosophila values across all samples. Primers are listed (5'-to-3') as follows:

| | |
|---|---|
| Pdea ChIP F | CTGCACACTTAGGGGTCTGT |
| Pdea ChIP R | CCCGCAATAGAGTCCTCCAT |
| Bai1 ChIP F | AACACGCACATTCTGTTCCC |
| Bai1 ChIP R | CAAGGAGGGGCTAGAGGATG |
| Ascl1 1 ChIP F | TCTAAGAAGTCTCCCGGGGA |
| Ascl1 1 ChIP R | GAACTTGGGTGCAGGAACAG |
| Ascl1 2 ChIP F | GGTCTCATCCTACTCGTCGG |
| Ascl1 2 ChIP R | GCTTCCAAAGTCCATTCGCA |
| Hes5 ChIP F | CCACCTAGTCTCTCTGGCAG |
| Hes5 ChIP R | GATCCTCAAGTCTGCCACCT |
| Olig1 ChIP F | CAGAAAGTGCTCGCTCTCAC |
| Olig1 ChIP R | AGGAAAAGAACCACCCCTCC |
| Olig2 ChIP F | CGTCTCAAGATCAACAGCCG |
| Olig2 ChIP R | CGTAGATCTCGCTCACCAGT |
| Kcnh2 ChIP F | TGATTTGAAGGGAAACGGCG |
| Kcnh2 ChIP R | TGGGGCCTTCAGATTGTCTT |
| Lingo1 ChIP F | CTCTTCTCAGCTGGTTCCCA |
| Lingo1 ChIP R | AGTGGAAGGAAGGGAGTTGG |
| Runx3 ChIP F | CTACGGGGATCTTTGGGGTGT |
| Runx3 ChIP R | AAGCTTCCAGGTGCCACTAT |
| Col20a1 ChIP F | GAGTGGGATCAGGAGCAGAG |
| Col20a1 ChIP R | ACCTGGCTTCTCTTCTGTCC |
| Drosophila Hoxa10 F | CAAGGCCTGTGTTTCTACGG |
| Drosophila Hoxa10 R | TTAAACTCAAGTCGCTGCCG |
| Drosophila Hoxa10-2 F | CCGTGTCTTAGCCCCGTAAT |
| Drosophila Hoxa10-2 R | TGCCTGCGTAACACTTGATG |
| Drosophila bcd F | GGCTCGAGTTCCCCATTTCA |
| Drosophila bcd R | GGGGAAGAAACAGGACGAGG |
| Drosophila croc F | CACGTTTGCAAGTCCATCGG |
| Drosophila croc R | CCGCACGCAGAATTCCTAGA |

**3'-Tag-seq and differential expression analysis.** RNA was extracted from cells using the NucleoSpin RNA Kit (Macherey-Nagel) and submitted to the University of California, Davis DNA Technologies Core. Libraries were prepared using the QuantSeq 3' mRNA-Seq Library Prep Kit FWD for Illumina (Lexogen), and sequencing was performed on an Illumina HiSeq4000. Two biological samples of each cell type were submitted for sequencing, and sequence alignment and differential expression analysis were performed using two established methodology pipelines developed for RNA-seq: STAR-EdgeR[44] and Tophat-Cufflinks[45]. Genes found to be differentially expressed by both methods were further analyzed via GOseq[46] (Bioconductor version 3.8) to identify overrepresented GO categories in comparing parental and gene-edited cells. GO analysis was not performed on the H3.3G34R lines due to the minimal overlap between HA-G34R and SF-G34R. Raw and analyzed 3'-Tag-seq data are accessible via GEO (accession series# GSE129765).

**siRNA knockdowns, drug treatments, and cell viability assays.** Cells were plated onto laminin-coated (10 μg/mL) plates 1 day prior to siRNA transfection or drug treatment. In the case of siRNA transfection, cells were transfected with one of two siRNAs targeting ASCL1 (Ambion #AM16704-41902 or #AM16704-114403) or a combination of both siRNAs or a single siRNA targeting RBPJ (Ambion #AM16708) according to the manufacturer's protocol using Genejuice transfection reagent (Millipore Sigma). Briefly, cells were plated in 6- or 96-well plates at a density of 200,000 and 6000 cells per well, respectively, 1 day prior to transfection. Seventy-five or 3 pmol of total siRNA was used along with 3 or 0.2 μL of Genejuice, respectively, for 6- or 96-well plates. siRNA transfection mixture was removed after 24 h, and a second transfection using the same reagent quantities was performed for an additional 24 h prior to assaying for cell viability. Noncoding siRNA (Ambion #AM4635) was used as negative control in the same quantities as the targeting siRNAs.

In the case of drug screening, cells were subjected to drug treatment for a period of 3 days. Cell viability was measured by using CellTiter Glo assay (Promega) and read on a SpectraMaxi3 spectrophotometer. Dose–response curves and IC50s were plotted and determined using the GraphPad Prism software. We first performed a drug treatment time course from 1 to 6 days using a saturating concentration of each drug based on the range of reported IC50s in the literature for various cell line treatments[17,29,47–50]. We confirmed that 3 days of treatment was sufficient to obtain significant differences in cell viability in a dose-dependent manner across our cell lines. DAPT and Panobinostat were purchased from SelleckChem, and RIN1 was purchased from AOBIOUS.

**Orthotopic xenograft assays.** Equal male-to-female ratios of NOD scid gamma immunodeficient mice were injected at 5–7 weeks old with cells from our panel of gene-edited lines via stereotactic injection into the pons to test for tumorigenicity and invasiveness of these cells. All protocols were approved by the UC Davis Institutional Animal Care and Use Committee and conformed to the institutional acceptable use policy. Mice were anesthetized with isoflurane (2–3% in oxygen) and placed into the stereotaxic frame. A small incision (1–2 cm) was made in the scalp to allow visualization of the skull and a small burr hole was drilled. In all, $1 \times 10^6$ glioma or control astrocyte cells were suspended in 2 μL of matrigel and injected using a needle and Hamilton syringe at a rate of 0.5 μL/min using coordinates that target the pons region via the IVth ventricle (0.5 mm lateral to the sagittal suture, 5.4 mm posterior to the bregma, and 3.1 mm deep). The incision was sutured, and Carprofen was administered by subcutaneous injection (5 mg/kg). Sutures were removed after 7 days. All mice were monitored daily for tumor formation or debilitation (>20% weight loss or other behavioral symptoms), or absence of such signs for a duration of 6 months, were euthanized. Upon reaching endpoint criteria, mice were anesthetized and perfused with 4% paraformaldehyde (PFA) prior to collecting brain tissue for histology. Brains were fixed in 4% PFA overnight and equilibrated sequentially to 15% and 30% sucrose followed by cryoembedding of tissue in OCT by submersion in dry ice and isopentane slurry.

**Tissue histology.** Histology sections were cut on a Licor Cryostat microtome and stored at −80 °C until used for staining. Tissue sections were stained using hematoxylin and eosin (H&E) using standard histology protocols (National Diagnostics). Histological analysis and scoring was performed in a single-blind manner in which samples were scored by a practicing pathologist without knowledge of the mutation status of the samples prior to scoring. H&E-stained samples were scored for number of tumors, tumor type, and morphology; estimated tumor size; level of atypia; vascular proliferation; tumor infiltration; and apoptosis. In addition, IF staining using anti-human nuclei (Millipore Sigma, #MAB1281B) and anti-H3K27M (Millipore Sigma, #ABE419) antibodies was performed on the tissue sections. Briefly, tissue sections were thawed and immersed in TBS with 0.25% Triton X-100 (wash buffer) for 5 min prior to blocking with 5% bovine serum albumin (BSA) in TBS with 0.25% Triton X-100 for 2 h at room temperature. Slides were drained, and primary antibody diluted in TBS with 2.5% BSA and 0.25% Triton X-100 was applied to the tissue overnight at 4 °C. Then the slides were rinsed 2× for 5 min with wash buffer followed by application of fluorophore-conjugated secondary antibody (AlexaFluor-546, goat anti-mouse, Thermo Fisher Scientific, #A11003) and incubated at room temperature for 1 h. After rinsing 3× for 5 min with PBS, slides were imaged on a Keyence BZ-X fluorescence microscope.

**Statistics and reproducibility.** Overall, pair-wise comparisons were made between H3.3 gene-edited cells lines that were otherwise isogenic. For statistics comparing global RNA and protein levels (by qPCR and Western blotting, respectively) or cell viability, Student's $t$ tests with a minimum of $n = 3$ biological replicates were performed and plotted using the GraphPad Prism software. ChIP-seq statistics was performed with the DiffBind program in R. 3'-Tag-seq differential statistics was analyzed via Cufflinks and EdgeR. Results described generally included a minimum of three biological replicates that are defined as independent biological samples either within an experiment or performed in subsequent experiments, except where explicitly stated otherwise.

**Reporting summary**. Further information on research design is available in the Nature Research Reporting Summary linked to this article.

## Data availability

All relevant data are available from the authors upon request. The sequencing datasets generated during and/or analyzed during the current study are available via GEO, accession series GSE129765. Other source data underlying plots shown in figures are provided in Supplementary Data 4.

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

## Acknowledgements

We thank Michelle Monje for advice and providing the DIPG-XIII, DIPG-XVII, and other cell lines. We thank David Allis for the SF188 cell line. We thank Maitreyee Jathal for the PCS2-HA-ASCL1 plasmid. We thank John McPherson for feedback on the manuscript. This work was supported by grants from the Alex Lemonade Stand Foundation, NINDS (1R01NS106878-01A1) and NIGMS (1R01GM116919) to P.S.K.

and by training fellowships to K.-Y.C. from the Oncogenic Signals and Chromosome Biology Postdoctoral Fellowship Program, National Cancer Institute (T32CA108459), and the Hartwell Foundation. A.M.C. acknowledges funding from ISCIII-FEDER (CP13/00189).

## Author contributions

P.S.K. and K.-Y.C. conceived of the project and designed the experiments. K.-Y.C. performed most of the experiments. K.B., R.H.K., V.C., N.L., and A.N. performed experiments. K.-Y.C., K.B., R.H.K., A.M.C, M.L., and P.S.K. analyzed the data. K.-Y.C. and P.S.K. wrote the paper.

## Competing interests

The authors declare no competing interests.
