## [Peer Review File · Communications Biology]

Reviewers' comments:

Reviewer #1 (Remarks to the Author):

Chen KY et al. seek to increase our understanding of the oncogenic role of histone H3.3 mutations found in pediatric gliomas. To identify major effectors the authors used CRISPR-Cas9 to introduce H3.3K27M and G34R mutations into previously H3.3-wildtype brain cells, while in parallel reverting the mutations in glioma cells back to wildtype. ChIP-seq analysis broadly linked K27M to altered H3K27me3 activity including specifically within super-enhancers, which exhibited perturbed transcriptional function. This was largely independent of H3.3 DNA binding. The K27M and G34R mutations induced several of the same pathways suggesting key shared oncogenic mechanisms including activation of neurogenesis and NOTCH pathway genes such as ASCL1. H3.3 mutant gliomas are also particularly sensitive to ASCL1 knockdown and to NOTCH pathway inhibition, which restored H3K27me3 levels at super-enhancer regions. Reciprocal editing of cells generally produced reciprocal effects on tumorigenicity in xenograft assays. Overall, their findings define common and distinct K27M and G34R oncogenic mechanisms, including potentially targetable pathways.

Overall these are interesting observations and I do believe they increase our understanding of these mutations. The focus on Notch as a pathway activated by the onco-histone mutations is novel. The notion that the two oncohistone mutations had some overlap in the pathways they activate is also novel. The head-to-head comparison between K27M H3.3 and G34R H3.3 mutations is interesting but one problem is that these normally arise in different parts of the brain (G34R mutant tumors are primarily hemispheric while K27M mutant tumors are primarily midline). This suggests a different cell-of-origin. So, one weakness of the study is that the intracranial injections were all done into the pons, a midline structure (including the G34R cells). It may be interesting to see what happens if some of the in vivo experiments were injected into the cerebral cortex in lieu of the pons. Also, a related caveat is the use of human brainstem astrocytes as a cell-of-origin. Again, the issue is that these are not the right cell-of-origin for G34R mutant gliomas (and probably not the right cell-of-origin for K27M mutant gliomas as these are thought to arise from OPCs). But with all of these caveats, I think this study does add to knowledge about these complex oncogenic events although most of the data is descriptive.

Reviewer #2 (Remarks to the Author):

The manuscript "Reciprocal H3.3 gene editing defines shared and distinct K27M and G34R mechanisms in pediatric glioma" by Chen et al. used CRISPR editing to generate isogenic cell lines with and without G34R and/or K27M mutations and assess the effect of the mutations on epigenetics, transcription and tumor formation. In two DIPG cell lines, XIII and XVII, they reverted the existing K27M mutation in H3F3A back to wild type and in a human astrocyte cell line (HA) and an H3 wild type glioma (SF188) they introduced K27M and G34R mutations into the endogenous H3F3A locus.

Unexpectedly, the authors show that the CRISPR mediated editing/correction of H3F3A produces changes in H3.3 protein levels (Supplemental Figure 3A and Supplemental Table 1), with introduction of the K27M mutation drastically increasing H3.3 protein levels and correction of the K27M mutation to wild type drastically reducing them. This change is not discussed explicitly by the authors although it is a feature of the model they present in Figure 7. This severely impairs the interpretation of their data, since the effect of the K27M mutation in particular cannot be untangled from the level of H3.3 expression.

Major Points:

1. The H3.3 protein levels reported are a major concern as they would impact interpretation of most of the experimental results presented. It is not clear however if the Western blot has even protein loading between samples. The authors report that the Western blot is on acid extracted histones, but the protein loading control used appears to be to beta-actin which should be only a contaminating component in an acid extraction and may vary between samples in a way that is not indicative of total histone protein. A more reliable check for protein loading would be using a total histone antibody to a histone that has not been altered experimentally; H4 for example.
2. There is no mention of spike in or bioinformatic methods used to normalize the ChIP-seq data. It has been established in the field (Pathania 2017, Mohammad 2017, Stafford 2018, Krug 2019, Harutyunyan 2019, Larson 2019, Silveira 2019, Nagaraja 2019) that these are needed for accurate assessment of ChIP-seq levels in situations with global changes in histone marks such as occur with H3K27me3 with the K27M mutation. The level of change in H3.3 protein suggests it may also be necessary for H3.3 ChIP-seq in the current study. This concern seems relevant from the DiffBind results presented in the paper showing five time more H3.3 peaks that are higher in WT than K27M than the reverse, despite the Western blot data showing that H3.3 protein is much higher in K27M than WT for each isogenic comparison.

If the H3.3 protein levels are being altered by CRISPR editing or correction of the K27M mutation as reported, the level of H3.3 protein and the K27M mutation are intertwined in such a way to make many of the results uninterpretable.

3. In Supplemental Figure 3, since H3K27me3, H3K27ac and H3K36me3 occur on all histone H3 isoforms, not just H3.3, normalization should be to total H3 levels, not H3.3. The legend says total H3, but no total H3 is shown in panel A.
4. Histone mark ChIP-seq should be normalized using spike in. The lack of normalization may be causing an overestimation of the number of H3K27me3 peaks higher in K27M than WT.

Reviewer #3 (Remarks to the Author):

General concerns

I have one main general concern, which is about novelty: two studies from Jabado group (Harutyunyan et al and Krug et al) and collaborators are cited but it is not discussed what the present study adds on their main findings. These studies are mostly based on cells that have been edited using Crispr/Cas. Probably the data on Notch are the most novel part of this study, but they are rather preliminary and require more work (further expanded below).

Main comments

- Fig S3: unclear why H3.3 levels are higher in the HA and SF lines upon editing of the gene to K27M. Also in the corrected lines, presence of the K27M mutation correlates with higher levels. This is rather confusing. The authors need to review the literature and come up with a potential explanation for this effect. Nonetheless, these differences are worrying with respect to interpretation of genomics data based on these lines.
- A second issue is the choice of astrocytes as control. I am not convinced there is evidence for astrocytes being the cell of origin of pHGG. I would rather use primary fetal NSCs. The authors should attempt gene editing in fNSCs, which are available for several lab and resources.

- Figure 2: differences in growth of edited line XVII point at oncogene addiction, but the XIII line does not show such differences. How is that? Authors need to come up with potential explanations (eg additional drivers in XIII etc)
- Figure 2: also the differences in adhesion should be explored in more detail. Is this because of differentiation? Careful analysis of differentiation markers should be employed. Increased adhesion in corrected XVII line correlates with decreased growth, suggesting that the two phenomena could be linked. Again, it is surprising that the authors did not set out to investigate this important phenotypic change.
- The similarities in effects of K27M and G34R is very interesting but not solid enough in my view.
- Figure 2: again, as mentioned above differences in H3.3 levels in cells carrying K27M make the interpretation of genomic data difficult.
- Figure 3: I am not sure I understand the rationale for comparing H3.3 and H3K27me3 genome distributions, as the authors cannot really discriminate from what is WT H3.3 and what is K27M H3.3 in mutant cells. I assume they want to infer something from changes in H3.3 distribution in edited cells, but it is rather an indirect way. Plus, as mentioned above there is the issue of higher H3.3 levels in cells carrying K27M H3.3. A more general concern for genomics run on these isogenic pairs is that XVII cells with the edited allele morphologically look very much different from edited XIII cells. Hence it is unclear whether observed changes at genomic level could in part represent indirect effects of different adhesion properties. This cannot be excluded and should be discussed extensively.
- Figure 4: data on overlap between differentially expressed genes in K27M and G34R edited cells is interesting, especially those related to Notch and neurogenesis. This remains of course quite descriptive, and the lack of G34R or G34V corrected lines (to WT) is a problem.
- Figure 5: data are interesting, ie the effect of DAPT. However, the authors need to control for overall K27me3 levels in cells treated with DAPT (WB on histone extracts). More generally, it is puzzling how reduction in Notch could affect H3K27me3 levels. The data on ASCL1 are also interesting, but again the potential mechanism here is unclear. Since siRNA works on these cells, the authors need to attempt this also for components of the Notch pathway, as this would provide clearer evidence that DAPT effects are specific (RBPJ Kd is an option or vice versa expression of Notch intracellular domains in corrected glioma cells). The latter is a critical experiment to strengthen the conclusions of this study.
- Figure 6 is the weakest figure of the paper, as data are not particularly convincing. A more detailed histopathological analysis should have been included. It is really unclear what are the cells carrying human markers in the transplants with corrected glioma cells. Authors need to run a complete histological characterization via IHC and IF. At present the figures shown in figure 6 do not allow the reviewers to fully judge the in vivo experiments. Also the SI contains panels that are very small and do not allow for any careful visual inspection.

Minor comments

- The end of introduction sound more like another abstract. It should be shortened to convey the main findings of the study.

Reviewers' comments:

Reviewer #1 (Remarks to the Author):

Chen KY et al. seek to increase our understanding of the oncogenic role of histone H3.3 mutations found in pediatric gliomas. To identify major effectors the authors used CRISPR-Cas9 to introduce H3.3K27M and G34R mutations into previously H3.3-wildtype brain cells, while in parallel reverting the mutations in glioma cells back to wildtype. ChIP-seq analysis broadly linked K27M to altered H3K27me3 activity including specifically within super-enhancers, which exhibited perturbed transcriptional function. This was largely independent of H3.3 DNA binding. The K27M and G34R mutations induced several of the same pathways suggesting key shared oncogenic mechanisms including activation of neurogenesis and NOTCH pathway genes such as ASCL1. H3.3 mutant gliomas are also particularly sensitive to ASCL1 knockdown and to NOTCH pathway inhibition, which restored H3K27me3 levels at super-enhancer regions. Reciprocal editing of cells generally produced reciprocal effects on tumorigenicity in xenograft assays. Overall, their findings define common and distinct K27M and G34R oncogenic mechanisms, including potentially targetable pathways.

Overall these are interesting observations and I do believe they increase our understanding of these mutations. The focus on Notch as a pathway activated by the onco-histone mutations is novel. The notion that the two oncohistone mutations had some overlap in the pathways they activate is also novel.

1. The head-to-head comparison between K27M H3.3 and G34R H3.3 mutations is interesting but one problem is that these normally arise in different parts of the brain (G34R mutant tumors are primarily hemispheric while K27M mutant tumors are primarily midline). This suggests a different cell-of-origin. So, one weakness of the study is that the intracranial injections were all done into the pons, a midline structure (including the G34R cells). It may be interesting to see what happens if the some of the in vivo experiments were injected into the cerebral cortex in lieu of the pons.

We thank the reviewer for their overall positive assessment of the study and their insightful suggestion to try xenografting the cell lines into the cerebral cortex given the varied locations of tumor formation in the brain. We are working on conducting more in vivo experiments to include different regions of injection, drug screening, and imaging. However, we feel this would be best suited as the focus of a separate study/manuscript. We have now mentioned in the present manuscript the caveat related to xenograft location relative to different locations of G34R and K27M tumors (Page 27).

2. Also, a related caveat is the use of human brainstem astrocytes as a cell-of-origin. Again, the issue is that these are not the right cell-of-origin for G34R mutant gliomas (and probably not the right cell-of-origin for K27M mutant gliomas as these are thought to arise from OPCs).

We definitely see the reviewer's point. In hindsight, this is something we wish we had started differently 4+ years ago. In fact, we have previously obtained human neural stem and progenitor cells (hNPCs) from Dr. Michelle Monje (Stanford). Four separate attempts over 18-24 months with various protocols at gene editing these cells all failed. While we could try again with additional different methods, it would require a minimum of six months and may not work. Even if successful, we would then need about another six months to repeat even just some of the key

assays (ChIP-seq, Tag-seq, ChIP-qPCR, WBs, drug studies, etc.). While we agree that neural stem/precursors may provide a somewhat better control, we feel the HAs are a reasonable choice as just one part of our overall reciprocal system. For comparison, a Nature Communications study (Harutyunyan, A. S. et al., 2019) used HEK293 cells as controls, and clearly HAs are much closer to the glioma cells and are normal cells. We additionally provided another 'gain-of-mutation' control in gene editing the SF188 glioma line to have the same H3.3 mutations, leading to many of the same downstream changes we see in the edited HA cells. Furthermore, the changes in HAs and SFs significantly overlap in a reciprocal manner with the changes observed with K27M back-mutated to WT in the 2 DIPG lines. Thus, our data overall strongly indicate that the changes we see in gene edited HA cells are relevant and important, even if HA cells are admittedly probably not an ideal control.

But with all of these caveats, I think this study does add to knowledge about these complex oncogenic events although most of the data is descriptive.

With this major revision, we have substantially increased the mechanistic insight into NOTCH signaling in the H3.3 mutant cell lines by targeting additional components of the pathway for knockdown or inhibition. In discussing these results, we feel that the data more convincingly supports real and relevant epigenetic perturbations leading to expression changes that are shared across multiple H3.3-mutated cell lines.

Reviewer #2 (Remarks to the Author):

The manuscript “Reciprocal H3.3 gene editing defines shared and distinct K27M and G34R mechanisms in pediatric glioma” by Chen et al. used CRISPR editing to generate isogenic cell lines with and without G34R and/or K27M mutations and assess the effect of the mutations on epigenetics, transcription and tumor formation. In two DIPG cell lines, XIII and XVII, they reverted the existing K27M mutation in H3F3A back to wild type and in a human astrocyte cell line (HA) and an H3 wild type glioma (SF188) they introduced K27M and G34R mutations into the endogenous H3F3A locus.

Unexpectedly, the authors show that the CRISPR mediated editing/correction of H3F3A produces changes in H3.3 protein levels (Supplemental Figure 3A and Supplemental Table 1), with introduction of the K27M mutation drastically increasing H3.3 protein levels and correction of the K27M mutation to wild type drastically reducing them. This change is not discussed explicitly by the authors although it is a feature of the model they present in Figure 7. This severely impairs the interpretation of their data, since the effect of the K27M mutation in particular cannot be untangled from the level of H3.3 expression.

We appreciate that changes in H3.3 or H3 levels may seem surprising given that H3 is often used as a ubiquitous control for histone expression in the literature, but we show that in the context of mutant H3.3, increased expression of H3.3 and thereby total H3, is a direct phenotypic consequence of the mutation. Though it was not included in our initial submission, we saw increased H3.3 expression in multiple H3.3K27M pediatric glioma cell lines even without CRISPR editing. This data is now included and discussed (Supplementary Fig. 1) in our revised manuscript. These H3.3 expression changes were confirmed by both Western blot and qPCR of *H3F3A* and *H3F3B*, where *H3F3B* expression was elevated in some cases in lines with higher H3.3 protein levels. One possible mechanism at another level is that H3.3 expression overall is increased in H3.3-mutated cells as a response to the dominant negative effect of mutant H3.3

preventing deposition of histone marks as well as impaired spread of H3K27me2/me3 marks along the chromatin, which was previously reported in (Harutyunyan, et al., 2019).

Thus, overall the increase total H3.3 protein levels tracking with the K27M (and sometimes G34R) mutations, and with gene edits of gain or loss of the glioma-associated mutations, is just one of several mutant phenotypes we define in this study, and by normalization including now by spike-in (more below) we can define the different phenotypes accurately.

Major Points:

1. The H3.3 protein levels reported are a major concern as they would impact interpretation of most of the experimental results presented. It is not clear however if the Western blot has even protein loading between samples. The authors report that the Western blot is on acid extracted histones, but the protein loading control used appears to be to beta-actin which should be only a contaminating component in an acid extraction and may vary between samples in a way that is not indicative of total histone protein. A more reliable check for protein loading would be using a total histone antibody to a histone that has not been altered experimentally; H4 for example.

All of our epigenetic data have been normalized to either total H3 or H3.3 levels when examining histone marks that are deposited on those histones. Comparison of numerous pairs of isogenic cell lines with only H3.3K27M mutation as well as examining histone expression across a large panel of glioma lines support that changing levels of H3.3 expression is a consequence of the mutation.

We agree with the reviewer that beta-actin is not a perfect loading control for acid extracts and have included H4 in our Western blots of acid extracts demonstrating the loading levels. We feel that changing H3.3 and H3 levels would not impair data interpretation so long as we normalize any examination of histone mark levels to the expression of H3.3 and total H3 in each cell lines (and in some cases to H4), which we have done for our Western blots and ChIP-qPCR data. In more than a dozen years of research studying H3.3, we have found in nuclear extracts that total H4 levels track tightly with beta-actin levels in the same extracts.

2. There is no mention of spike in or bioinformatic methods used to normalize the ChIP-seq data. It has been established in the field (Pathania 2017, Mohammad 2017, Stafford 2018, Krug 2019, Harutyunyan 2019, Larson 2019, Silveira 2019, Nagaraja 2019) that these are needed for accurate assessment of ChIP-seq levels in situations with global changes in histone marks such as occur with H3K27me3 with the K27M mutation. The level of change in H3.3 protein suggests it may also be necessary for H3.3 ChIP-seq in the current study. This concern seems relevant from the DiffBind results presented in the paper showing five time more H3.3 peaks that are higher in WT than K27M than the reverse, despite the Western blot data showing that H3.3 protein is much higher in K27M than WT for each isogenic comparison.

While spike-in normalization may have become more common in glioma studies, it is not considered necessary in most ChIP-Seq studies and is not specifically recommended by ENCODE. Spike-in normalization with Drosophila chromatin tends to identify additional regions that have (lower) changes in levels that may be missed due to differences in antibody to protein ratios (e.g. between WT and K27M samples), but large changes such as those we identified should still be consistent. Nonetheless, we have now performed spike-in ChIP-qPCR using Drosophila chromatin and show that with and without normalization, the large-scale epigenetic

changes we observe are extremely consistent, exemplified by spike-in ChIP-qPCR analysis of the select gene regions we identified by ChIP-Seq to undergo these changes when mutating H3.3. While it's true that the H3.3 ChIP-seq peaks and H3.3 Western blot quantification levels do not exhibit precisely the same kinds of changes, ChIP-seq measures chromatin-bound H3.3, while acid extract Western blots measure all H3.3 molecules in the nucleus, whether bound to chromatin or not, so differences can arise from this distinction.

If the H3.3 protein levels are being altered by CRISPR editing or correction of the K27M mutation as reported, the level of H3.3 protein and the K27M mutation are intertwined in such a way to make many of the results uninterpretable.

While we can appreciate that changes in H3.3 protein levels include the changes in mutant H3.3K27M as well as wildtype H3.3, we believe that with normalization to the total H3 and H3.3 levels for each cell line, the changes in histone marks we report as well as the differences in chromatin deposition of these histones and marks, especially when considering changes that are universal to multiple reciprocally edited pairs of isogenic cells, provide valuable and interpretable insights into biological changes as a direct consequence of H3.3 mutation.

3. In Supplemental Figure 3, since H3K27me3, H3K27ac and H3K36me3 occur on all histone H3 isoforms, not just H3.3, normalization should be to total H3 levels, not H3.3. The legend says total H3, but no total H3 is shown in panel A.

We thank the reviewer for catching this error in our figure image. We now have included total H3 as well as H4 in our figure showing the corresponding normalizations. The figure legend was correct in that we normalized H3K27me3, H3K27ac, and H3K36me3 to total H3 levels, while H3.3S31p was normalized to H3.3 levels.

4. Histone mark ChIP-seq should be normalized using spike in. The lack of normalization may be causing an overestimation of the number of H3K27me3 peaks higher in K27M than WT.

The principle of normalizing to spike-in chromatin is to account for differences in antibody or chromatin loading, but generally corrects or underestimates peaks in regions with less binding (Orlando, et al., Cell Reports). As we mentioned in response to point #2, we now perform spike-in normalization ChIP followed by qPCR on a subset of gene regions and show that with and without the normalization, the changes in binding we report in our study, which are the largest changes observed by ChIP-seq and confirmed by ChIP-qPCR in our cell lines, remain highly consistent and do not change the conclusions.

Reviewer #3 (Remarks to the Author):

General concerns

I have one main general concern, which is about novelty: two studies from Jabado group (Harutyunyan et al and Krug et al) and collaborators are cited but it is not discussed what the present study adds on their main findings.

The methodological similarities and differences between our study and the Jabado group studies as well as the novelty of our study was discussed in the Discussion and additionally analyzed by clustering our data with their data in Supp Fig 4E (Now Supp Fig 5C). Our study is distinct and novel in several ways: 1) The Jabado group used CRISPR to knockout the mutant

H3.3 allele while we corrected or introduced the H3.3 point mutations via CRISPR. 2) Jabado group used HEK293 cells as a control line for inducing H3.3K27M, whereas we utilized human astrocytes and SF188 glioma cells. 3) Importantly, we included the study of cell lines mutated to have G34R, and thus were able to compare mechanisms of K27M and G34R mutations. These distinctions translate to some interesting and important differences between our work and theirs, particularly in H3K27me3 deposition (Supp Fig 5C). These data support that while some consequences remain similar between knockout versus point mutation of a particular gene, the two modes of intervention can lead to distinct changes in downstream effectors and how we interpret the process of gliomagenesis. Finally, we believe our system of precise gene editing changing single codons is more elegant and less likely to lead to extraneous effects.

These studies are mostly based on cells that have been edited using Crispr/Cas. Probably the data on Notch are the most novel part of this study, but they are rather preliminary and require more work (further expanded below).

We have addressed this with substantial new data and detail the interesting new findings related to Notch below.

Main comments

- Fig S3: unclear why H3.3 levels are higher in the HA and SF lines upon editing of the gene to K27M. Also in the corrected lines, presence of the K27M mutation correlates with higher levels. This is rather confusing. The authors need to review the literature and come up with a potential explanation for this effect. Nonetheless, these differences are worrying with respect to interpretation of genomics data based on these lines.

Given the similarity of the reviewer's concern with that of Reviewer 2, we would ask the reviewer to please refer to our responses to Reviewer 2's initial summary and their point #1.

- A second issue is the choice of astrocytes as control. I am not convinced there is evidence for astrocytes being the cell of origin of pHGG. I would rather use primary fetal NSCs. The authors should attempt gene editing in fNSCs, which are available for several lab and resources.

Given the similarity of the reviewer's concern with that of Reviewer 1, we would ask the reviewer to please refer to our response to Reviewer 1's point #2. Briefly, we have previously obtained human neural stem and progenitor cells (hNPCs), but four separate attempts with various protocols at gene editing these cells all failed due in large part to cell viability issues post-editing. For comparison, a Nature Communications study (Harutyunyan, A. S. et al., 2019) used HEK293 cells as controls, and clearly HAs are closer to the glioma cells and are normal cells. We additionally provided another 'gain-of-mutation' control in gene editing the SF188 glioma line to have the same H3.3 mutations, leading to many of the same downstream changes we see in the edited HA cells.

- Figure 2: differences in growth of edited line XVII point at oncogene addiction, but the XIII line does not show such differences. How is that? Authors need to come up with potential explanations (eg additional drivers in XIII etc)

We thank the reviewer for their insight and agree that cell line phenotypic differences and additional driver mutations are an important aspect of the gliomas. We now have included expanded discussion of these differences in the Discussion (Page 24).

- Figure 2: also the differences in adhesion should be explored in more detail. Is this because of differentiation? Careful analysis of differentiation markers should be employed. Increased adhesion in corrected XVII line correlates with decreased growth, suggesting that the two phenomena could be linked. Again, it is surprising that the authors did not set out to investigate this important phenotypic change.

We share the reviewer's interest in cell adhesion. However, while additional manipulation either by gene editing or drug inhibition of adhesion effectors may be an interesting pursuit for future study, we feel such work is beyond the scope of our current study. Although the adhesion properties of the XVII and XIII cells do change and correlate with changes in expression of cell junction and cell-cell signaling genes (Fig 4 and deposited Tag-seq data), the gene expression changes we report are observed reciprocally when adding or removing the H3.3 K27M mutation by gene editing (not just in XVII and XIII, but also when adding K27M or G34R to astrocyte or SF188 lines). It is possible that some genomic changes such as altered gene expression occur downstream of cell adhesion regulatory changes. However, we feel that given the consistency of these changes that occur in a reciprocal manner in reciprocally-edited cell lines, they are likely arising due to changes in H3.3 mutation status and are not due to unintended changes due to the process of gene editing (also please recall we saw no off-target gene editing effects) and cell culture.

- The similarities in effects of K27M and G34R is very interesting but not solid enough in my view.

We would respectfully argue that introducing K27M and G34R mutations precisely by gene editing into both of exactly the same 2 cell lines (SF and HA), and then analyzing phenotypic changes is a highly robust way to define and compare potential mechanisms, and superior to other possible routes such as exogenous high-level overexpression of transgenes. Unfortunately, as detailed more below, we have as yet been unsuccessful at CRISPR reversion of naturally occurring G34R mutations back to WT in patient cells and don't see prospects of being able to successfully do so any time soon, which admittedly would have added further strength to the study.

- Figure 2: again, as mentioned above differences in H3.3 levels in cells carrying K27M make the interpretation of genomic data difficult.

The regions where our analysis found consistent H3K27me3 changes specifically do not exhibit significant changes in H3.3 levels as shown by overlapping our H3K27me3 and H3.3 ChIP-seq data. While global levels of H3.3 are elevated in K27M lines, this seems to be an important consequence of regulatory changes driven by H3.3 mutation. We feel that epigenomic data should be interpretable given we are able to also examine corresponding H3.3 ChIP data for each genomic region (and normalize if needed). As mentioned in response to Reviewer #2, we have also in a sense normalized and validated our ChIP-Seq data by *Drosophila* chromatin spike-in within ChIP-qPCR analysis.

- Figure 3: I am not sure I understand the rationale for comparing H3.3 and H3K27me3 genome distributions, as the authors cannot really discriminate from what is WT H3.3 and what is K27M H3.3 in mutant cells. I assume they want to infer something from changes in H3.3 distribution in edited cells, but it is rather an indirect way. Plus, as mentioned above there is the issue of higher H3.3 levels in cells carrying K27M H3.3. A more general concern for genomics run on these isogenic pairs is that XVII cells with the

edited allele morphologically look very much different from edited XIII cells. Hence it is unclear whether observed changes at genomic level could in part represent indirect effects of different adhesion properties. This cannot be excluded and should be discussed extensively.

We agree that the morphological and adhesion changes associated with H3.3 mutation status are quite interesting. In response to the reviewer's suggestion, we have now much more extensively discussed this and its potential ties to our genomic data in the revised Discussion section (Page 26). To some extent, it is kind of a 'chick-or-egg' situation since one cannot readily determine whether adhesion/morphological changes or genomic changes occur first. In either case, they are highly relevant to tumorigenesis, and the relationship may be better defined in future studies.

- Figure 4: data on overlap between differentially expressed genes in K27M and G34R edited cells is interesting, especially those related to Notch and neurogenesis. This remains of course quite descriptive, and the lack of G34R or G34V corrected lines (to WT) is a problem.

G34R-mutant glioma lines are much rarer than K27M-mutant lines. While we have been able to obtain cryovials of 2 such cell lines, neither could not be propagated long term. Thus far our attempts at CRISPR editing the few G34R glioma lines we were able to obtain and grow even for some weeks have been unsuccessful due to poor cell viability. We have added additional experiments perturbing the NOTCH (and overlapping neurogenesis) pathways to strengthen our conclusions (please see response to the next point).

- Figure 5: data are interesting, ie the effect of DAPT. However, the authors need to control for overall K27me3 levels in cells treated with DAPT (WB on histone extracts). More generally, it is puzzling how reduction in Notch could affect H3K27me3 levels. The data on ASCL1 are also interesting, but again the potential mechanism here is unclear. Since siRNA works on these cells, the authors need to attempt this also for components of the Notch pathway, as this would provide clearer evidence that DAPT effects are specific (RBPJ Kd is an option or vice versa expression of Notch intracellular domains in corrected glioma cells). The latter is a critical experiment to strengthen the conclusions of this study.

In addition to siRNA knockdown of ASCL1, we now include siRNA knockdown of RBPJ (downstream of NOTCH signaling) as well as drug inhibition of RBPJ with RIN1, leading to similar reduction in cell viability as treatment of DAPT. We agree with the reviewer that reduction of NOTCH leading to changes in H3K27me3 levels is surprising, but also intriguing. The mechanism by which this may occur is so far mostly unclear to us as well given that the NOTCH pathway components do not include any epigenetic modifiers. We did not observe the same changes in global H3K27me3 levels with RIN1 and therefore have removed the panel showing changes in H3K27me3 by CHIP-qPCR post-DAPT treatment from this study as we are working on additional drug screening studies for which we believe the probing of the link between NOTCH signaling and epigenetic modification could be more suited.

- Figure 6 is the weakest figure of the paper, as data are not particularly convincing. A more detailed histopathological analysis should have been included. It is really unclear what are the cells carrying human markers in the transplants with corrected glioma cells. Authors need to run a complete histological characterization via IHC and IF. At present the figures shown in figure 6 do not allow the reviewers to fully judge the in vivo

experiments. Also the SI contains panels that are very small and do not allow for any careful visual inspection.

An extensive histological characterization of xenograft tissue was performed by an experienced cancer pathologist on our team. We now focus on the IHC data in addition to our IF images showing cell engraftment and tumor formation. The main xenograft figure has been improved with higher magnification H&E images, and the complete histology scoring table is now included as a supplementary table.

Minor comments

- The end of introduction sound more like another abstract. It should be shortened to convey the main findings of the study.

The introduction has been condensed as suggested.

REVIEWERS' COMMENTS:

Reviewer #1 (Remarks to the Author):

The revised manuscript is significantly improved. The authors did a nice job addressing my critiques raised in the initial review.

This work does add to our understanding of histone mutant gliomas.

The novelty lies in the comparison between K27M and G34R mutations in an isogenic system. The focus on the Notch pathway downstream of the oncohistones is also relatively novel.

The resources developed will be of value to the scientific community.

Reviewer #3 (Remarks to the Author):

The authors have conducted extensive work to address my comments. I particularly appreciated efforts to investigate the role of Notch signaling using genetic tools rather than simply pharmacological agents. This the main novel finding of the study. However, I remain unconvinced about the (lack of) explanation regarding the higher H3 and H3.3 levels in cells carrying the mutant histone. These changes represent at the same time a technical hurdle for genomics and a potentially interesting biological effect. In both cases, they should have been investigated more in depth.